# Large and irreversible future decline of the Greenland ice-sheet

Jonathan M. Gregory[1,2,*], Steven E. George[1,*], and Robin S. Smith[1,*]

[1]National Centre for Atmospheric Science, University of Reading, UK
[2]Met Office Hadley Centre, Exeter, UK
[*]All authors made essential contributions to this work.
**Correspondence:** Jonathan Gregory (j.m.gregory@reading.ac.uk)

**Abstract.**

We have studied the evolution of the Greenland ice-sheet under a range of constant climates typical of those projected for the end of the present century, using a dynamical ice-sheet model (Glimmer) coupled to an atmosphere general circulation model (FAMOUS–ice AGCM). The ice-sheet surface mass balance (SMB) is simulated within the AGCM by a multilayer snow scheme from snowfall and surface energy fluxes, including refreezing and dependence on altitude within AGCM gridboxes. Over millennia under any warmer climate, the ice-sheet reaches a new steady state, whose mass is correlated with the magnitude of global climate change imposed. If a climate that gives the recently observed SMB were maintained, GMSLR would reach 0.5–2.5 m. For any global warming exceeding 3 K, the contribution to GMSLR exceeds 5 m. For the largest global warming considered (about +5 K), the rate of GMSLR is initially 2.7 mm yr$^{-1}$, and eventually only a small ice-cap endures, resulting in over 7 m of GMSLR. Our analysis gives a qualitatively different impression from previous work, in that we do not find a sharp threshold warming that divides scenarios in which the ice-sheet suffers little reduction from those in which it is mostly lost. The final steady state is achieved by withdrawal from the coast in some places, and a tendency for increasing SMB due to enhancement of cloudiness and snowfall over the remaining ice-sheet by the effects of topographic change on atmospheric circulation, outweighing the tendency for decreasing SMB from the reduction of surface altitude. If late 20th-century climate is restored after the ice-sheet mass has fallen below a threshold of about 4 m of sea-level equivalent, it will not regrow to its present extent, because the snowfall in the northern part of the island is reduced once the ice-sheet retreats from there. In that case, about 2 m of GMSLR would become irreversible. In order to avoid this outcome, anthropogenic climate change must be reversed before the ice-sheet has declined to the threshold mass, which would be reached in about 600 years at the highest rate of mass-loss within the likely range of the Fifth Assessment Report of the Intergovernmental Panel on Climate Change.

# 1 Introduction

## 1.1 Mass-loss from the Greenland ice-sheet in recent decades

During 1961–1990 the Greenland ice-sheet had a roughly constant mass, in which snowfall was balanced by the sum of surface ablation (meaning all processes of mass-loss, predominantly liquid runoff due to melting) and solid discharge of ice into the sea (forming icebergs). Over the last 30 years both ablation and discharge have increased significantly while snowfall has not (Shepherd et al., 2012; van den Broeke et al., 2016; Bamber et al., 2018; Mouginot et al., 2019). In recent years, the mass-loss from the Greenland ice-sheet of $239 \pm 20 \, \mathrm{Gt \, yr^{-1}}$ (in 2012–2017, Shepherd et al., 2020), or about $0.7 \, \mathrm{mm \, yr^{-1}}$ sea level equivalent (SLE), accounts for about 20% of global-mean sea-level rise (GMSLR), most of which is due to thermal expansion of seawater (*i.e.* thermosteric) or mass-loss from glaciers.

The increase in discharge is probably the ice-dynamical response of outlet glaciers to reduced buttressing by their ice-tongues, which have thinned due to basal melting by warmer sea-water (Holland et al., 2008). The increase in ablation causes 60% of the mass-loss (van den Broeke et al., 2016; Fettweis et al., 2017). It has been partly due to anthropogenic climatic warming, which is amplified at high northern latitudes, and partly to recent unusual atmospheric circulation (Tedesco et al., 2013; Fettweis et al., 2017; Pattyn et al., 2018; Trusel et al., 2018). In recent years, the surface mass balance $S = P - R$, where $P$ is snowfall and $R$ ablation, has fallen lower than during the warm period in Greenland in the early 20th century (Fettweis et al., 2017) and summer temperatures have risen higher (Hanna et al., 2012). Some recent summers have seen surface melting over practically the entire ice-sheet because of high air temperature, decreased cloudiness and reduction of albedo, the latter due to the increase of snow grain size and the exposure of bare ice, both caused by surface snow melting (Tedesco et al., 2013, 2016; Hofer et al., 2017; Trusel et al., 2018).

## 1.2 Projections of future mass-loss

The future of the Greenland ice-sheet is one of the large uncertainties in projections of GMSLR (Church et al., 2013; Clark et al., 2016). Ice discharge is projected to increase in coming decades with rising water temperature, but it will decline on longer timescales as the ice-sheet thins at the coast and its outlet glacier termini retreat inland (Nick et al., 2013; Fürst et al., 2015; Aschwanden et al., 2019). On multicentennial timescales, SMB is dominant and the source of greater uncertainty (Fürst et al., 2015).

Projections indicate that ablation will increase non-linearly with temperature and more rapidly than snowfall, meaning that SMB will continue to decline and the rate of mass-loss will grow, especially under scenarios of high $CO_2$ emissions (Gregory and Huybrechts, 2006; Fettweis et al., 2013; Vizcaíno et al., 2014; Pattyn et al., 2018; Rückamp et al., 2018; Golledge et al., 2019; Aschwanden et al., 2019). Recent projections of the contribution of the Greenland ice-sheet to GMSLR mostly lie within the likely ranges of the Fifth Assessment Report (AR5) of the Intergovernmental Panel on Climate Change (Church et al., 2013) *viz.* 0.04–0.12 m and 0.09-0.28 m by 2100 relative to 1986–2005 under scenarios RCP2.6 and RCP8.5 respectively. The range of uncertainty arises from the spread in global warming simulated by atmosphere–ocean general circulation models (AOGCMs)

and in their amplification of warming in Greenland relative to global warming, as well as the sensitivity of Greenland SMB to regional climate change (Gregory and Huybrechts, 2006; Fettweis et al., 2013).

Although substantial, the contribution from the Greenland ice-sheet is only 10–30% of projected GMSLR by 2100. Its importance is greater on multicentury timescales, because its size (mass $M = 7.4$ m SLE) implies a large commitment to GMSLR. Thinning of the ice-sheet due to increasing ablation is affected by a positive feedback loop between SMB and elevation: as the surface elevation falls, the surface air temperature rises, and surface melting increases, magnifying the ablation increase. We refer to this as the local lapse-rate feedback. Another positive feedback on ablation is caused by the decrease in surface albedo due to melting, as in recent years (Tedesco et al., 2016). Despite these feedbacks, a steady state could be regained with an ice-sheet of smaller mass but little loss of area if the reduction of SMB were compensated by the reduction of discharge resulting from thinning of outlet glaciers (Rückamp et al., 2018) (see Section 1.3). The reduction of mass would be mitigated if snowfall increases, which is projected by AOGCMs.

On the other hand, previous work indicates there may be a threshold $T_c$ of global-mean surface air temperature change $\Delta$SAT (relative to pre-industrial) beyond which the ice-sheet will be greatly reduced or vanish entirely (Huybrechts et al., 1991; Gregory et al., 2004; Gregory and Huybrechts, 2006; Robinson et al., 2012) (see Section 1.3). Levermann et al. (2013) estimated $T_c = 0.8$–2.2 K, using the model of Robinson et al. (2012) constrained by information from the last interglacial (Robinson et al., 2011). If this range is correct, limiting $\Delta$SAT to 2.0 K in accordance with the Paris agreement or to its aspiration of 1.5 K could make a critical difference to whether $T_c$ is exceeded (Pattyn et al., 2018). Loss of the Greenland ice-sheet would cause much greater GMSLR than from glacier mass-loss or thermosteric sea-level rise for similar degrees of warming (Church et al., 2013; Levermann et al., 2013) although, even in the most extreme scenarios, the complete removal of the ice-sheet would take a least a thousand years (*e.g.* Ridley et al., 2005; Aschwanden et al., 2019).

### 1.3  Discussion of the threshold warming

The rate of change of the mass of the ice-sheet is $dM/dt = S - D$, where $D$ is discharge. In the unperturbed steady state $dM/dt = 0 \Rightarrow S = D$ *i.e.* SMB is balanced by discharge. In a warmer climate, ablation $R$ and snowfall $P$ both increase, but $\Delta R > \Delta P \Rightarrow \Delta S = \Delta P - \Delta R < 0$ (see references in Section 1.2), where $\Delta$ denotes the difference from the initial state. A new steady state can be achieved if $\Delta D = \Delta S$ *i.e.* if discharge reduces by as much as SMB, so that $D + \Delta D = S + \Delta S$.

Let us suppose that raising the global-mean SAT by $T$ initially perturbs the SMB by an amount $\Delta S_T(T) < 0$. Further suppose that a new steady state can be achieved, with little change in ice-sheet area, in which discharge is reduced by marginal thinning, such that $\Delta D = \Delta S = \Delta S_T(T)$. The larger $T$, the greater the reduction in discharge needed to balance $\Delta S_T(T)$. The threshold $T = T_c$ is reached when there is just sufficient marginal thinning to cause all outlet glaciers to retreat from the coast, reducing discharge $D + \Delta D$ to zero. To attain a balance, SMB must also fall to zero, with $\Delta S = -S$. Hence $\Delta S_T(T_c) = -S$ defines $T_c$.

Any $T$ exceeding $T_c$ will give negative SMB, but discharge cannot be further reduced (*i.e.* below zero) to compensate. The unbalanced negative SMB will reduce the thickness of the ice-sheet and the altitude of the surface, making the SMB even more negative by the local lapse-rate feedback. If no other process is involved, the ice-sheet will be completely eliminated for any $T > T_c$ by this feedback loop, which is called the "small ice-cap instability". The threshold has been estimated as $T_c = 1.9$–

4.5 K relative to pre-industrial (Gregory and Huybrechts, 2006; Meehl et al., 2007), by evaluating the warming required to reduce SMB to zero with the present-day surface topography. The same method gave $T_c = 2.1$–$4.1$ K in the AR5 (Church et al., 2013, Section 13.4.3.3).

Robinson et al. (2012) showed that this method of calculation overestimates the actual $T_c$ for onset of the small ice-cap instability. One possible contribution to the difference is the reduction in SMB, neglected above, due to the local lapse-rate effect of the marginal thinning *before* the threshold is reached. Let us write this contribution to $\Delta S$ as $\Delta S_L$. For steady state at the threshold, we now require $\Delta S_T(T_c) + \Delta S_L = -S \Rightarrow \Delta S_T(T_c) = -(S + \Delta S_L)$. Since $\Delta S_L < 0$, the right-hand side is less negative than before, so $T_c$ is smaller.

An ice-sheet model is required to allow for $\Delta S_L$ in quantifying $T_c$, because the change in ice-sheet topography is determined simultaneously by SMB change and ice-dynamical change. With their model, Robinson et al. (2012) demonstrated that SMB may initially be positive but decline to zero as the topography changes, whereupon the instability is triggered, leading to the eventual loss of the ice-sheet. They determined their lower $T_c$ by finding the final steady state for various $T$ and versions of their coupled climate–ice-sheet model, and our approach is similar. The coupled system simulated by their and our models is 100 considerably more complicated than this simplified conceptual treatment, which is intended to illustrate the idea. The actual outcome is affected by further feedbacks, both positive and negative, which we discuss later.

## 1.4   Possibility of irreversible mass-loss

If the ice-sheet were removed then, even after $CO_2$ fell and global climate returned to pre-industrial, it might not be possible to regenerate it, because of greater ablation or reduced snowfall due to lower elevation and albedo in deglaciated regions (Toniazzo 105 et al., 2004). If the ice-sheet did not regrow, it would imply that its pre-industrial steady state is a relict of a colder climate (Solgaard et al., 2013). Previous work shows there may be more than two steady states for pre-industrial climate (Charbit et al., 2008; Ridley et al., 2010; Solgaard and Langen, 2012; Robinson et al., 2012). Stable states of intermediate size (between zero and present-day) are possible because of the interaction of the ice-sheet with its own climate through atmospheric dynamics, whereby its surface topography affects regional precipitation and temperature, like mountains do. The existence of intermediate 110 states means that partial loss of the ice-sheet could be irreversible.

    It is the possibility of threshold behaviour (*i.e.* "tipping-points") and irreversibility which makes the future of the Greenland ice-sheet of particular concern (Pattyn et al., 2018). Precautionary action to mitigate the threat of irreversible damage is a principle of the Framework Convention of Climate Change (Article 3.3), even when there is not full scientific certainty. The serious implications of the uncertainty are the motivation for the work presented in this paper, in which we reexamine the 115 future decline and possible recovery of the ice-sheet. Our conclusions differ in some critical ways from those of previous work, because of the greater complexity of the model, which we next describe.

## 2 Model

Previous work on the subject has used simplified climate models, or a small set of climate states, or been limited to a few centuries into the future. In the present work we use a dynamic ice-sheet model coupled to a atmosphere general circulation model (AGCM) to study the transient and steady states of the ice-sheet over tens of millennia. Typical AGCMs are not suitable for modelling ice-sheet SMB, because they do not have adequate treatments of albedo and hydrology, nor fine enough spatial resolution for the large gradients in topography and climate parameters across the margins of the ice-sheets, where much of the snowfall and snowmelt occurs (Vizcaíno, 2014). Specially developed regional climate models (RCMs) have proven very useful for high-resolution projections and process studies (*e.g.* MAR RCM, Fettweis et al., 2013; RACMO RCM, Noël et al., 2018) but they require lateral boundary conditions (BCs) from global AGCMs, and cannot feed back on climate change outside their domain. Moreover, computational expense prevents the use of these RCMs in studying ice-sheet evolution over millennia. The first such experiment, with MAR coupled to an ice-sheet model, was only 150 years long (Le clec'h et al., 2019). In multimillennial studies, empirical parametrisations for SMB as a function of surface air temperature (*e.g.* Reeh, 1989), precipitation etc. have often been applied. Being calibrated for observed climate, such schemes may be less reliable for simulations of very different climates of the future or past, and when used in coupling to an AGCM they imply surface energy and water fluxes which are unrelated to those within the AGCM, thus violating conservation.

### 2.1 FAMOUS–ice AGCM

For sufficient speed, we use the FAMOUS AGCM, which is the atmosphere component of the FAMOUS AOGCM (Smith et al., 2008; Smith, 2012), itself a low-resolution version, at 7.5° longitude by 5° latitude, of the HadCM3 AOGCM (Gordon et al., 2000). For physical consistency, we calculate the SMB in the AGCM, but Greenland spans only seven gridboxes in longitude and five in latitude in the free atmosphere of the model, which is far from adequate for simulating the important effects of topographic gradients and snow hydrology for ice-sheets. Therefore in this work we use "FAMOUS–ice", a new version of FAMOUS (version xotzb, Smith et al., submitted), incorporating a multilayer surface snow scheme which calculates melting, refreezing of meltwater, runoff and SMB on "tiles" at a set of elevations within each AGCM gridbox (each tile covering a fraction of the gridbox area). This is similar to the method implemented for the Greenland ice-sheet in the Community Earth System Model (CESM) (Vizcaíno et al., 2013; Lipscomb et al., 2013; Muntjewerf et al., 2020) and we use the same ten elevations. Smith et al. show the improvement in the cumulative distribution of area as a function of altitude (the hypsometry) that results from the subgridscale treatment. Below we summarise the FAMOUS–ice SMB and coupling schemes, of which further details are given by Smith et al.

For vertical interpolation of atmospheric variables from the AGCM gridbox elevation to the tile elevations we prescribe a lapse rate of $6\,\mathrm{K\,km^{-1}}$ for air temperature. This we obtained from the climate of 1980–1999 simulated by Fettweis et al. (2013) with the MAR RCM using sea-surface BCs (sea-surface temperature and sea-ice) from MIROC5, the AOGCM which Fettweis et al. found to give the most satisfactory SMB simulation. Downwelling longwave radiation and specific humidity are vertically interpolated in FAMOUS–ice using gradients consistent with the prescribed lapse rate, but precipitation is not

redistributed vertically, nor modified in phase. The same uniform air temperature lapse rate for Greenland is used *e.g.* by Aschwanden et al., 2019, and found by Sellevold et al. (2019) to give the most similar SMB gradient to RACMO in their CESM ice-sheet coupling, which, like our scheme, does not downscale precipitation.

We have paid particular attention to the treatment of the surface albedo of the Greenland ice-sheet, to which SMB is very sensitive. Bare ice has lower albedo than snow in FAMOUS–ice and snow albedo has different values for visible and near-
155 infrared, both dependent on the snow-grain size, which is a prognostic that depends on the "ageing" of the surface snow by melting and refreezing following new snowfall. There is an uncertain parameter in the relationship between snow-grain size and albedo. In our experiments, we use three alternative parameter values that are consistent with observations of albedo. For convenience we refer to these as low, medium and high albedo, but the reader should keep in mind that the albedo is variable in each case. More details are given by Smith et al. (submitted).

Instead of simulating sea surface conditions by using the FAMOUS AOGCM, we use the AGCM alone, for both recent and future climate, with sea-surface BCs derived from AOGCM experiments of the Coupled Model Intercomparison Project Phase 5 (CMIP5) and atmospheric $CO_2$ concentration to give the corresponding radiative forcing (see Table 2 and the start of Section 3). We use the AGCM for two reasons. First, the FAMOUS AOGCM has larger biases in its simulation of recent climate than MIROC5 and the three other AOGCMs we use (CanESM2, HadGEM2-ES and NorESM1-M), which have all
previously been selected as satisfactory for Greenland regional climate simulation (see Fettweis et al., 2013, and van Angelen et al., 2013, for evaluation of their regional climate simulations). Second, this method allows us to investigate the uncertainty in Greenland ice-sheet projections that arises from the spread of climate projections given by AOGCMs for any given scenario. The AGCM sea-surface BCs are 20-year climatological monthly means, which lack interannual variability; we have checked that statistically indistinguishable results for the ice-sheet are obtained with the AGCM cycling through a 20-year series of
monthly mean BCs for the same climate (Figure S1a).

By prescribing sea surface conditions, we exclude any climate interaction between the ice-sheet and the ocean, in particular, possible cooling of regional climate due to weakening of the AMOC caused by meltwater from the ice-sheet (*e.g.* Vizcaíno et al., 2010). There is wide uncertainty in this aspect of ocean climate change, whose implications for the ice-sheet could possibly be explored in further work by modifying the sea-surface temperatures in a range of ways to represent the effects of
175 AMOC changes projected by AOGCMs (Stouffer et al., 2006; Gregory et al., 2016).

## 2.2 FiG coupling and spinup

We use the Glimmer-CISM community ice-sheet model (Rutt et al., 2009, `https://cism.github.io`) with the shallow-ice approximation at 20 km grid-spacing and no basal sliding. Consequently the model does not simulate ice-streams or rapid ice-sheet dynamics, and it will inevitably underestimate the rate of ice-sheet mass-loss, especially in coming decades. This is
180 acceptable because our aim is not to make realistic time-dependent projections, but to study the steady state obtained under constant climates.

Because the ice-sheet model lacks sufficient resolution and physical processes to simulate calving into fjords, we instantly remove ice which flows beyond the present margin of the ice-sheet. This BC prevents a tendency for the ice-sheet to expand

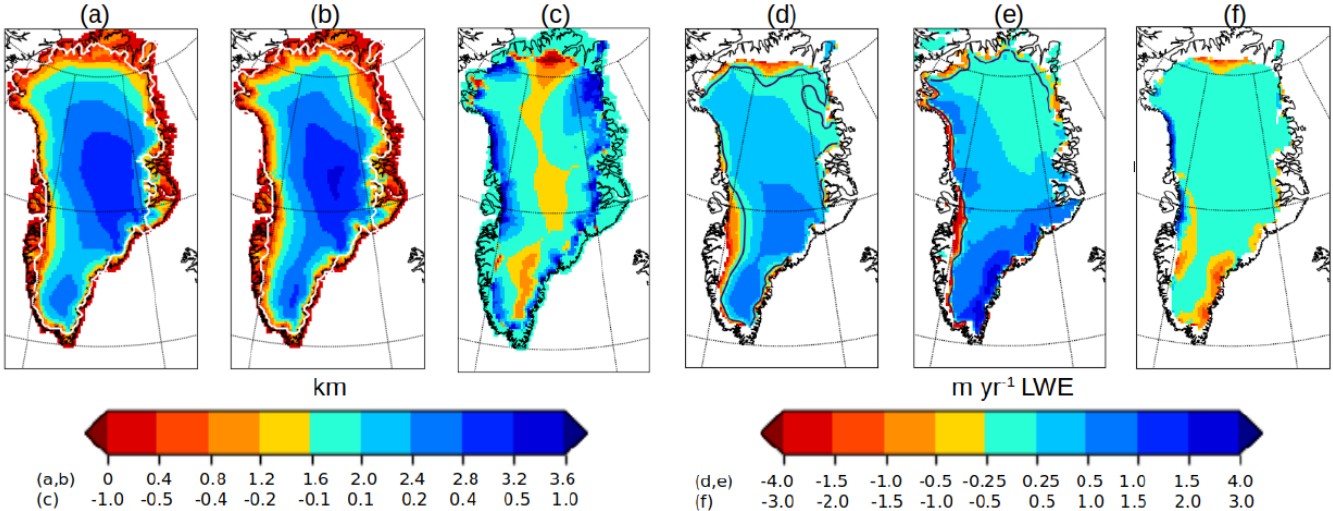

**Figure 1.** (a,b) Greenland surface elevation above sea level in (a) FAMOUS–ice (medium albedo) and (b) observations (Bamber et al., 2001a, b). The white contour is the observed ice margin, the same in both maps. (c) Difference between (a) and (b), positive means FAMOUS–ice surface is higher. (d,e) Specific surface mass balance (expressed as liquid water equivalent) for the climate of MIROC5 1980–1999 in (d) FAMOUS–ice (medium albedo), (e) MAR. The black contour is the equilibrium line (where specific SMB is zero). (f) Difference between (d) and (e), positive means FAMOUS–ice SMB is more positive.

slightly, and thus it makes the modelled ice edge coincide with the observed one. It becomes irrelevant in most of our experiments, when the ice-sheet contracts. For simplicity in the model we omit isostatic uplift, which in reality gives a negative feedback on ice-sheet mass-loss through the local lapse-rate feedback, because it is not a large effect (*e.g.* 2% over 1000 years, Aschwanden et al., 2019) and does not seem necessary given that our scenarios are idealised in other ways as well.

The AGCM and the ice-sheet model are coupled to make FAMOUS–ice–Glimmer (FiG, Gregory et al., 2012; Roberts et al., 2014; Smith et al., submitted). After each AGCM year, the SMB simulated by the AGCM is interpolated horizontally and vertically (with the AGCM tile elevation as the vertical coordinate) to the ice-sheet surface topography, and the AGCM topography and land-surface properties are updated according to the ice-sheet model. When the ice-sheet retreats, the newly exposed land is assigned the properties of bare soil, including a low snow-free albedo; its properties do not subsequently change because vegetation dynamics are not included in the model.

FiG runs at about 220 simulated AGCM years per wallclock day on six cores, with the AGCM consuming the great majority of the CPU time. Although this is fast for an AGCM, it is not fast enough for multimillennial experiments. Therefore, after each AGCM year, the ice-sheet model runs for ten years with the resulting SMB field, depending on the assumption that the local lapse-rate feedback will be negligible for changes in topography that occur within that decade, before the AGCM runs again. We have verified that this 10:1 acceleration makes no significant difference to our results (Figure S1a). Hereafter by "year" in FiG experiments we mean an ice-sheet year except where otherwise stated.

Because our aim is to simulate ice-sheet response to climate change over millennia, we have to start from a coupled steady state, with little long-term tendency in the ice-sheet topography. We initiate the ice-sheet model with observed topography (Bamber et al., 2001a, b) and run FiG under the MIROC5 AOGCM climate of 1980–1999, during which period the ice-sheet was near to a steady state in reality (van den Broeke et al., 2016). In the first millennium the ice-sheet mass $M$ increases by 0.1–0.2 m SLE. With medium and low albedo it subsequently decreases again more slowly, while with high albedo it continues to grow slowly and stabilises after 4 kyr at 0.3 m SLE above present-day (Figure S1b). The states obtained after about 4 kyr of spin-up are used to initiate the experiments described in Section 3. In these states $M$ is close to reality, and the topography similar to observed (Figure 1a,b,c), with the summit and southern dome altitudes being a few 100 m too low. (See also Appendix B concerning the constraint implied on albedo by the requirement of realistic $M$.)

## 2.3 Simulated surface mass balance for recent climate

Comparing the three choices of albedo in FAMOUS–ice with BCs for the MIROC5 1980–1999 climate, we find that lower albedo produces lower SMB (the first group of cases in Table 1 differ significantly at the 10% level), because ablation is greater due to greater snowmelt, but snowfall is about the same (slightly larger with higher albedo because of greater ice-sheet area). For the same albedo (medium), the SMB is significantly lower with the CanESM2 and NorESM1-M historical climates than with MIROC5 because the ablation is larger, whereas the SMB is about the same with HadGEM2-ES as with MIROC5 (the second group in the table). This shows the influence of the different climate simulations of the AOGCMs.

A similarly large spread in SMB arises from the choice of Greenland model (FAMOUS–ice, MAR or RACMO), both because they simulate somewhat different regional climate in the free atmosphere and over land when given climate BCs from the same AOGCM, and because they have different SMB schemes. MAR has much larger SMB than FAMOUS–ice with the MIROC5 climate, because of smaller ablation, while RACMO has larger ablation than FAMOUS–ice with the HadGEM2-ES climate (the third group in the table). Comparison with MAR and RACMO for ERA-interim BCs (*i.e.* observationally derived, the fourth group in the table) suggests that FAMOUS–ice with high albedo is similar to both of them.

Regarding its geographical distribution, FAMOUS–ice SMB interpolated to the Glimmer grid compares favourably with the MAR simulation for the MIROC5 climate (Figure 1d,e,f). It shows positive and negative values of realistic magnitude, and reproduces the important geographical features, including the confinement of negative SMB to the margins, especially on the west coast, the decrease in positive SMB towards the north-east, and the occurrence of greatest positive SMB in the strip of maximum snowfall along the south-east coast. We presume that the latter is not sufficiently intense in FAMOUS–ice because of the low resolution of the AGCM. The equilibrium line (black contour) is generally a little higher and further inland in FAMOUS–ice (see Smith et al., submitted, for details).

## 3 Mass-loss of the ice-sheet in warmer climates

We run a set of 47 FiG experiments to study the SMB change ($\Delta$SMB), rate of mass-loss and eventual steady state of the Greenland ice-sheet, using the three different choices of FAMOUS–ice snow-albedo parameter, with 20-year climatological

| | Climate | Greenland model (albedo) | SMB | Snowfall | Ablation |
|---|---|---|---|---|---|
| | **1980-1999 climates** | | | | |
| 1 | MIROC5 | FAMOUS–ice (low) | $310 \pm 10$ | 693 | 383 |
| | MIROC5 | FAMOUS–ice (medium) | $332 \pm 11$ | 697 | 364 |
| | MIROC5 | FAMOUS–ice (high) | $414 \pm 9$ | 715 | 300 |
| 2 | CanESM2 | FAMOUS–ice (medium) | $272 \pm 21$ | 681 | 409 |
| | HadGEM2-ES | FAMOUS–ice (medium) | $312 \pm 20$ | 705 | 393 |
| | NorESM1-M | FAMOUS–ice (medium) | $287 \pm 16$ | 721 | 434 |
| 3 | MIROC5 | MAR | $437 \pm 24$ | 681 | 244 |
| | CanESM2 | MAR | $410 \pm 23$ | 635 | 225 |
| | HadGEM2-ES | RACMO | $244 \pm 25$ | 660 | 416 |
| | NorESM1-M | MAR | $483 \pm 16$ | 691 | 208 |
| 4 | ERA-interim | MAR | $388 \pm 23$ | 637 | 249 |
| | ERA-interim | RACMO | $406 \pm 22$ | 683 | 277 |
| | **MIROC5 2080-2099 climates** | | | | |
| 5 | RCP2.6 | FAMOUS–ice (medium) | $325 \pm 14$ | 704 | 379 |
| | RCP4.5 | FAMOUS–ice (medium) | $150 \pm 25$ | 735 | 585 |
| | RCP8.5 | FAMOUS–ice (medium) | $-207 \pm 35$ | 805 | 1013 |
| | **Mean over AOGCM climates** | | | | |
| 6 | 1980–1999 | FAMOUS–ice (medium) | 307 | 703 | 395 |
| | 2080–2099 RCP2.6 | FAMOUS–ice (medium) | 212 | 746 | 533 |
| | 2080–2099 RCP4.5 | FAMOUS–ice (medium) | 60 | 777 | 716 |
| | 2080–2099 RCP8.5 | FAMOUS–ice (medium) | −273 | 825 | 1098 |

**Table 1.** Greenland area-integral surface mass balance (SMB), snowfall and ablation (all in $\mathrm{Gt\,yr^{-1}}$) for FAMOUS–ice with MIROC5 AOGCM historical climate (with the three choices of FAMOUS–ice albedo), FAMOUS–ice with historical climates of other AOGCMs (FAMOUS–ice medium albedo only), the MAR and RACMO RCMs with the same AOGCM climates and with ERA-interim climate (from Table 2 of Fettweis et al., 2013), FAMOUS–ice with MIROC5 AOGCM climate (medium albedo only) under RCP scenarios, and the FAMOUS–ice (medium albedo) mean over available AOGCMs for each climate (no HadGEM2-ES for RCP8.5, all four AOGCMs in other cases). The first column identifies the "groups" of results into which the table is divided; we refer to these group numbers in the text. Ablation is SMB − snowfall, mainly runoff from snowelt, and including evaporation, sublimation, condensation and rainfall freezing in the snowpack (in the RCMs; all rainfall runs off in FAMOUS–ice). The ± uncertainty shown for SMB is the standard error of the time-mean, estimated by assuming annual values to be independent. The SMB from FAMOUS–ice has smaller standard errors than from the RCMs for two reasons. First, the FAMOUS–ice simulations exclude interannual variability due to SST and sea-ice by using climatological mean BCs. Second, the RCM time-means use 20 years of data, while we use 100 years for the FAMOUS–ice MIROC5 1980–1999 simulations, which supply our initial steady states, and 30 for other FAMOUS–ice simulations, which are transient states.

| CMIP5 scenario | years | $CO_2$ | ERF | notes |
|---|---|---|---|---|
| historical | 1980-1999 | 402 | 2.1 | |
| RCP2.6 | 2080–2099 | 402 | 2.1 | |
| RCP4.5 | 2080-2099 | 650 | 4.7 | |
| RCP8.5 | 2080–2099 | 1200 | 8.0 | not with HadGEM2-ES |
| abrupt4xCO2 | 121–140 | 1200 | 8.0 | CanESM2 and low albedo only |
| abrupt4xCO2 | 101–120 | 1200 | 8.0 | HadGEM2-ES and low albedo only |

**Table 2.** AOGCM climates used to supply sea-surface boundary conditions for the first set of FiG experiments. The BCs mostly determine the climate, with only a relatively small influence from the $CO_2$ concentration (in ppm). This is "equivalent $CO_2$", chosen for RCP4.5 and RCP8.5 to give approximately the nominal effective radiative forcing in RCPs at 2100 (ERF, $W\,m^{-2}$), with all other forcing agents kept as pre-industrial. For simplicity, regarding 1980–1999 as "present day", we decided to use the same concentration for historical and RCP2.6 simulations. We consider this acceptable because the AR5 median assessment of the net anthropogenic ERF in 2011 is $2.3\,W\,m^{-2}$, with a likely range of $1.1$–$3.3\,W\,m^{-2}$, and the difference between this and the nominal forcing of $2.6\,W\,m^{-2}$ under RCP2.6 at 2100 is small compared with the large systematic uncertainty. Similarly for simplicity we used the same $CO_2$ concentration for RCP8.5 and abrupt4xCO2.

monthly mean sea-surface BCs taken from the four selected CMIP5 AOGCMs for five climate scenarios (Table 2). These five are the late 20th century (1980–1999, called "historical"), the end of the 21st century under three RCP scenarios (representative concentration pathway, as in the AR5; van Vuuren et al., 2011), and quadrupled pre-industrial $CO_2$ (abrupt4xCO2, warmer than any RCP). The experiments have steady-state climates. This is unrealistic, but it simplifies the comparison, and is reasonable since no-one can tell how climate will change over millennia into the future. Our simulations should be regarded only as indicative, rather than as projections. Each experiment begins from the FiG spun-up state for MIROC5 historical climate with the appropriate albedo parameter. Although in most cases there is a substantial instantaneous change in BCs when the experiment begins, the land and atmosphere require only a couple of years to adjust.

### 3.1 Evolution of surface mass balance

Our set of BCs produces a wide range of global mean surface air temperature change $\Delta$SAT of $-1$ to $+5$ K, relative to the MIROC5 historical climate. Some are negative because the historical climate is warmer in MIROC5 than in the other three AOGCMs. In warmer climates, snowfall and ablation are both increased (Table 1, fifth group shows results with MIROC5 RCP climates, last group shows the mean over results for each of the available AOGCMs for each climate). In general, the greater the global warming, the more negative the $\Delta$SMB initially produced, relative to the time-mean MIROC5 historical state with the same albedo (Figure 2a). For a given scenario, the AOGCMs give a range of $\Delta$SAT, as is very well-known (*e.g.* Collins et al., 2013). In our set of AOGCMs, NorESM1-M warms least, HadGEM2-ES most (Figure S2a). $\Delta$SAT in FAMOUS–ice and the BCs is very highly correlated (Figure S2b). The spread of FAMOUS–ice results with BCs from different AOGCMs for a given warming is due to their different relationships between global $\Delta$SAT and Greenland regional climate change (shown by the grey lines in Figure S2c).

Global warming under RCP2.6 is fairly small, leading to small ΔSMB, especially for MIROC5 (squares near 1.0 K in Figure 2a), although MIROC5 is in the middle of the range for RCP8.5 (squares near 3.5 K). For mean over AOGCM climates under RCP2.6, RCP4.5 and RCP8.5, FAMOUS–ice with medium albedo gives SMB change of of $-95$, $-247$ and $-580$ Gt yr$^{-1}$ respectively with respect to the mean over AOGCM historical climates.

The greatest global warming is given by HadGEM2-ES abrupt4xCO2. With low albedo, this climate produces the most negative SMB of $-756$ Gt yr$^{-1}$ in the time-mean of the first 300 years, during which the topography change from the initial state is still quite small (Figure 3a1,b1). It is also the most negative ΔSMB of $-1066$ Gt yr$^{-1}$ relative to the MIROC5 historical climate with low albedo. Although this is a large ΔSMB, that of Aschwanden et al. (2019) for RCP8.5 is larger still, perhaps because they use a degree-day scheme and assume geographically uniform warming.

In our experiment, the specific SMB is strongly negative all around the margin and especially in the southern dome, where it has a local maximum in the historical climate (Figure 3a2,b2). The snowfall on the ice-sheet is ∼10% larger in the abrupt4xCO2 climate (Figures 3a4,b4). We note that the precipitation is ∼50% larger, consistent with the warming in Greenland of 11 K and the increase of ∼5% K$^{-1}$ found by previous studies *e.g.* Gregory and Huybrechts (2006), but the snow fraction declines from ∼90% to ∼70%. The downwelling surface shortwave radiation in summer (June–August) is smaller because cloudiness

is greater (Figures 3a3,b3). Both the increased snowfall and the reduced insolation tend to make ΔSMB positive, but ΔSMB is actually large and negative because of the overwhelming effect of increased downwelling surface longwave radiation, which is mainly due to the air above the ice-sheet being warmer, and partly to the increase in cloud cover.

We find that the relationship between ΔSAT and ΔSMB in the set of FiG experiments roughly follows the cubic formula (shown as the solid curve in Figure 2a) derived by Fettweis et al. (2013) for MAR projections and used in the AR5 for the

---

**Figure 2 *(following page).*** Relationships between various quantities in the first set of experiments, with FiG under constant climates listed in Table 2, and run to a steady state, as shown in Figure 4b. All panels use the key of (a) for colours; (b–e) use the key of (a) for symbols. (a) Time-mean ΔSMB *vs.* ΔSAT, both for the first 300 years relative to the initial steady state under the historical MIROC5 climate with the same albedo parameter. The solid curve is the cubic relationship fitted by Fettweis et al. (2013) to MAR projections, and the dashed curves delimit the likely range of the AR5. (b) Trajectories of ice-sheet SMB (not ΔSMB) *vs.* mass $M$, shown as 200-year means for the first millennium, and 1000-year means thereafter. The trajectories begin at the symbols, with $M$ close to the observed for the present day, and a wide range of SMB. They end with a wide range of $M$, but all have positive SMB. (c) Final steady-state $M$ *vs.* time-mean ΔSAT in FAMOUS–ice for the first 300 years. (d) Final steady-state $M$ of the ice-sheet *vs.* time-mean ΔSMB of the first 300 years. The vertical dashed lines mark the observational estimates of ΔSMB for the recent periods and studies shown in the key (van den Broeke et al., 2016; Mouginot et al., 2019; Shepherd et al., 2020); for van den Broeke et al. we used the steady-state SMB for 1961–1990 and the SMB trend for 1991–2015. The oblique solid and dashed lines are linear regressions of $M$ *vs.* ΔSMB and vice-versa respectively for ΔSMB $> -700$ Gt yr$^{-1}$. The solid horizontal lines indicate the threshold of irreversibility for medium and low albedo, and the solid vertical lines translate them into ΔSMB thresholds, with uncertainty ($\pm 2$ standard deviations) shown by the grey band. (e) Trajectories of ice-sheet thickness (volume divided by area) *vs.* specific SMB for 1000-year means, beginning at the symbols. (f) Trajectories of $M$ *vs.* ice-sheet area as grey lines, with the final configurations indicated by the symbols, and the fitted power-law relationship shown by the black line.

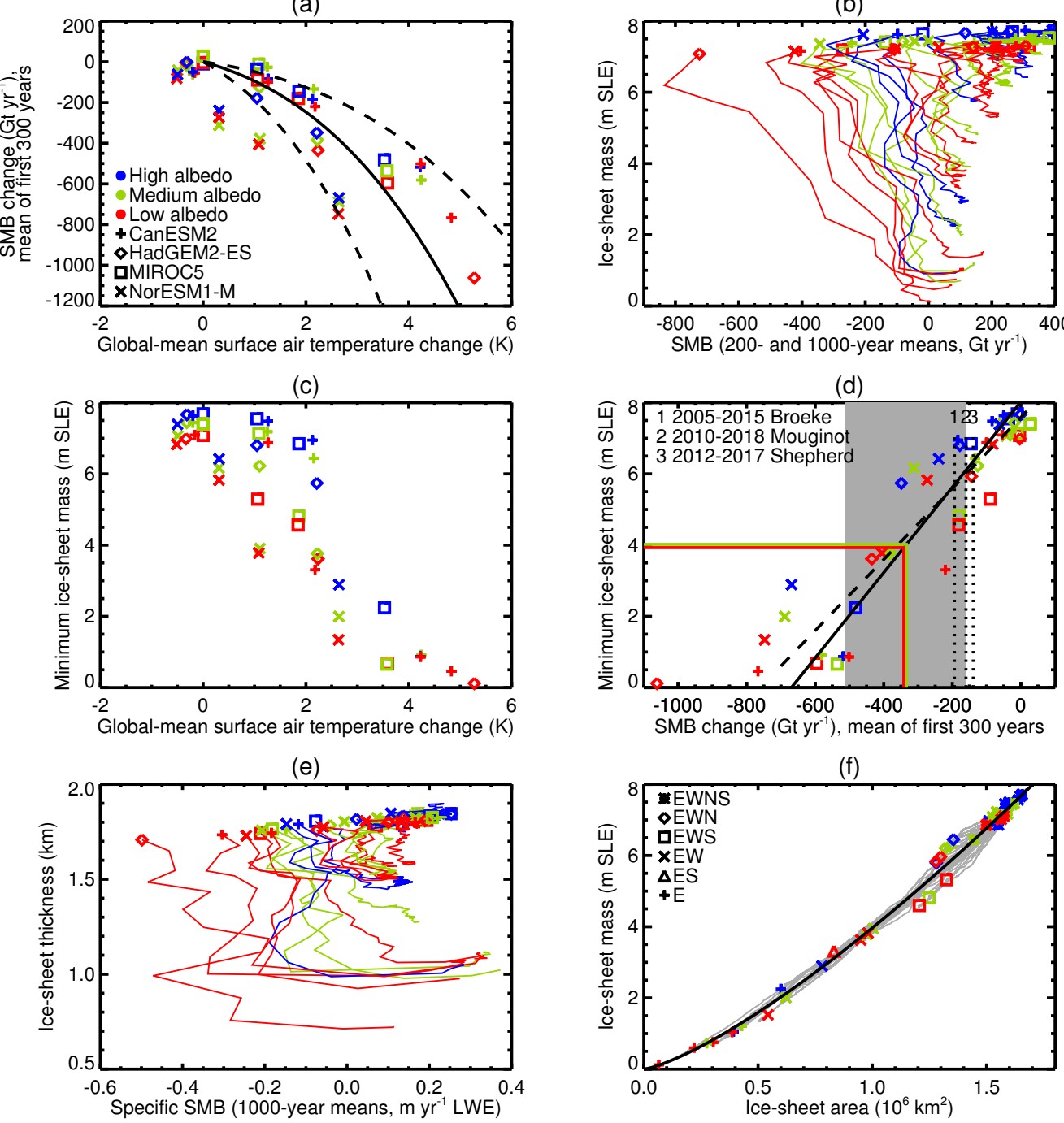

Greenland contribution to GMSLR. There is a small spread due to the choice of albedo parameter, and a larger spread due to choice of AOGCM. The FiG $\Delta$SMB mostly lies within the AR5 likely range (dashed curves). In the majority of cases SMB remains positive (Figure 2b), but because the ice-sheet was initially in balance, negative $\Delta$SMB leads to loss of mass (Figure 4a). In the most extreme case, rapid retreat of the ice-sheet margin reduces the discharge by a third in the first century alone; this slightly offsets $\Delta$SMB, giving ice-sheet mass-loss of 2.5 mm yr$^{-1}$ SLE.

Because of the effect of lowering topography, the SMB becomes more negative in most cases during the early centuries (Figure 2b). For the the 21st century, this effect is omitted in our experiments, since we instantaneously impose the climates from the end of the century on the initial state. This is an acceptable approximation because the effect is small on that timescale *e.g.* Edwards et al. (2014) give a best estimate of 4.3% for the consequent increment in the GMSLR contribution by 2100, but this increases with time, *e.g.* to 9.3% by 2150 (Le clec'h et al., 2019), and 9.6% by 2200 (Edwards et al., 2014). In the

28 cases with $\Delta$SMB $< -100$ Gt yr$^{-1}$ in the time-mean of the first century of our experiments, $\Delta$SMB becomes about 20% more negative on average during the second and third centuries due to the local lapse-rate feedback, about twice the size of the effect estimated by Edwards et al.. Thereafter the SMB becomes gradually more positive again (Figure 2b), because the area contracts, with the areas most prone to ablation being removed most quickly, as happens with a retreating mountain glacier.

## 3.2   Final ice-sheet mass and global-mean sea-level change

The experiments continue until the ice-sheet reaches a steady state (defined as $|dM/dt| < 0.02$ mm yr$^{-1}$ SLE over 2000 years). The longest experiments, which take 40 kyr (Figure 4b), are for large climate change (RCP8.5), which entails a large loss of mass, with high albedo, which causes a relatively slow rate of mass-loss. The shortest are the experiments in which a different historical climate from MIROC5 is applied, because the effect on SMB of differences among AOGCMs in their simulations of late 20th-century climate is relatively minor.

There is a wide range of $M$ in the final steady state (Figures 2b and 4b), between slightly greater than present-day (in some historical experiments), and almost zero (in abrupt4xCO2). With one exception, historical and RCP2.6 climates produce final $M$ of 6 m SLE or more (implying GMSLR not exceeding 1.5 m), while RCP8.5 climates all produce final $M$ of 3 m SLE or less

---

**Figure 3** *(following page).* Illustrative states of the ice-sheet, all from coupled FAMOUS–ice–Glimmer experiments except for column (d), as follows: (a) initial state with HadGEM2-ES historical climate and low albedo; (b) initial state with HadGEM2-ES abrupt4xCO2 climate and low albedo; (c) transient state from the experiment of (b); (d) transient state of uncoupled Glimmer with the climate and 3D SMB of (b); (e) final state with CanESM2 abrupt4xCO2 climate and low albedo; (f) final state with MIROC5 RCP4.5 climate and medium albedo; (g,h) final states with MIROC5 historical climate and low albedo, regrown from transient states with $M = 3.83$ m SLE and $M = 4.03$ m SLE respectively in the experiment of (b). The quantity shown in each row in colours, and by contour lines in rows (3–4), is stated above its colour bar. Row (1) is an instantaneous state; rows (2–4) are time-means of 30 FAMOUS–ice years, equivalent to 300 FiG years. The ice-sheet edge is shown by a thick black line in rows (2–4). The numbers in the bottom right corner are in row (1) ice-sheet mass in m SLE, (2) ice-sheet area-integral SMB in Gt yr$^{-1}$, (4) ice-sheet area-integral snowfall in Gt yr$^{-1}$. The symbols in row (1) indicate steady-state configurations by the key of Figure 2f.

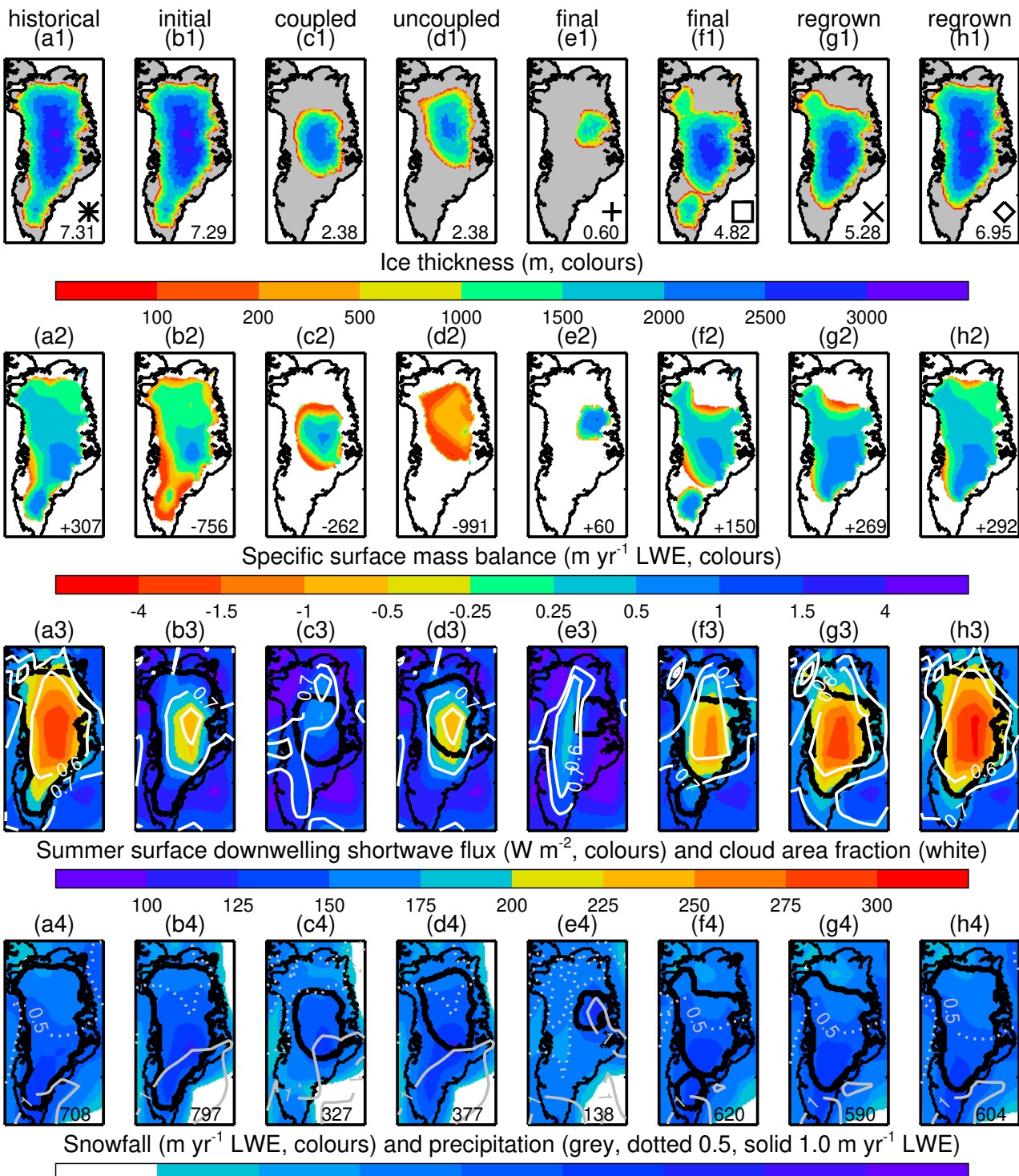

(GMSLR exceeding 4 m). In all cases the SMB is finally positive (Figure 2b), and must be balanced by ice discharge, meaning that the ice-sheet does not retreat entirely inland.

There is a clear tendency for climates of greater $\Delta$SAT to produce smaller ice-sheets, but the final $M$ has quite a wide range for any given initial global-mean annual-mean $\Delta$SAT within 1–4 K (Figure 2c). For given BCs, we have found in test experiments that the ice-sheet evolution follows somewhat different trajectories from slightly different initial states, but that they converge on very similar final states (Figure S1a). Thus, the scatter in Figure 2c is not random noise, but arises from the detailed interaction of the evolving ice-sheet topography with its regional climate, which depends on the choice of BCs. The final $M$ depends on which AOGCM is used, because of their different patterns of SST and sea-ice change; this dependence is omitted if the warming is assumed to be uniform (*e.g.* Robinson et al., 2012; Aschwanden et al., 2019).

The Pearson product–moment correlation coefficient between final $M$ and initial $\Delta$SAT is $-0.89$, and the Spearman rank correlation coefficient $-0.83$. The correlation is similar if for $\Delta$SAT we use Greenland area-mean summer-mean air temperature change, either at the surface or at 600 hPa (the latter as Fettweis et al., 2013) (Figure S2d,e,f). However, the relationship is better-defined using $\Delta$SMB instead of $\Delta$SAT (Figure 2d), with both product–moment and rank correlation coefficients of 0.92. If the initial $\Delta$SMB is near zero, the ice-sheet changes little; the more negative the initial $\Delta$SMB, the smaller the final $M$. Excluding the case with the most negative $\Delta$SMB, a linear relationship is a fairly good fit.

With any choice of albedo, for any $T > 3$ K, the final steady-state $M \lesssim 2$ m SLE, meaning GMSLR exceeds 5 m. It is important to note, however, that the spread of final $M$ does not suggest a sharply defined threshold in $T$ beyond which a complete or nearly complete loss of the ice-sheet ensues (Figure 2c). For low and medium albedo, there is a fairly monotonic decline in size of the steady state from near present-day $M \simeq 7$ m at $T = 0$ K to $M < 1$ m for $T > 3$ K. For high albedo, there might be a transition from $M \simeq 6$ m at $T = 2.0$ K to $M \simeq 3$ m at $T = 2.5$ K—to obtain a clearer description of the behaviour, more experiments are needed in this part of the diagram. In any case, the interval between temperatures giving a "large" and a "small" final ice-sheet is wider in our results, or alternatively the mass interval between "large" and "small" is narrower, than in the results of Levermann et al. (2013) (their Figure 1C). All of the versions of their model have a sharp transition between $M > 6$ m and $M < 1$ m over a temperature interval which appears to be less than 0.1 K. Our model gives a qualitatively different impression of the transition.

If negative feedbacks were neglected, there would be no final ice-sheet for negative initial SMB, as described in Section 1.3. Actually all final states have positive SMB and non-zero $M$, although some have initially negative SMB (Figure 2b). In our model, if *any* climate warmer than historical is maintained indefinitely the ice-sheet will contract to a new non-zero steady state, whose size depends on the magnitude of the warming and the consequent SMB perturbation.

Observational analyses indicate that recent $\Delta$SMB (with respect to a steady state before the 1990s) is between $-200$ and $-150$ Gt yr$^{-1}$, with substantial interannual variation (*e.g.* van den Broeke et al., 2016; Mouginot et al., 2019; Shepherd et al., 2020; dotted lines in Figure 2d). If a climate giving such a $\Delta$SMB were maintained it would eventually lead to GMSLR of 0.5–2.5 m according to the linear fit (Figure 2d, allowing for the range of FiG initial $M$). On the basis of the MAR simulations of Fettweis et al. (2013), the CMIP5-mean projection of $\Delta$SMB for 2080–2099 climate is $-242$ Gt yr$^{-1}$ under RCP4.5 and $-710$ Gt yr$^{-1}$ under RCP8.5. According to the fit, the former implies eventual GMSLR of about 3 m, the latter about 7 m.

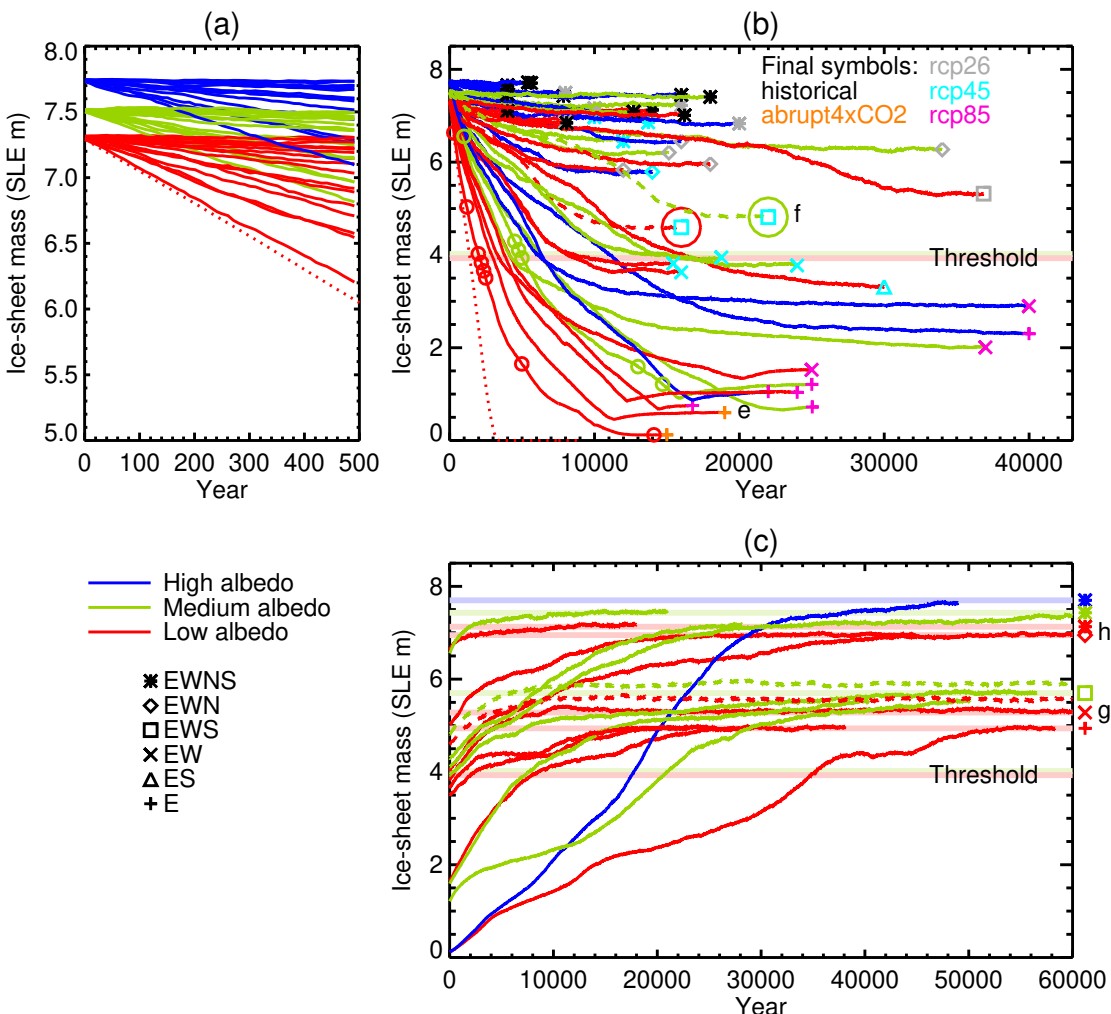

**Figure 4.** Timeseries of Greenland ice-sheet mass with constant climates. (a,b) First set of experiments, beginning from steady states for MIROC5 historical (1980–1999) climate, and continuing until a new steady state is reached under the scenarios indicated by the colours of the final symbols in (b) according to the final symbol colour key in that panel. The solid and dashed lines are FiG experiments; the dotted line is the experiment with the uncoupled Glimmer ice-sheet model. The circles indicate transient and final states which provide the initial states for the second set of experiments. (c) Second set of FiG experiments, beginning from states of the same albedo, and continuing until a new steady state is reached under under the MIROC5 historical climate. The single high-albedo experiment begins from the low-albedo initial state of smallest mass. The experiments shown by dashed lines in (c) begin from the final states of the experiments shown by dashed lines in (b). In all panels, the FAMOUS–ice albedo is indicated by the line colours. In (b) and (c), the final symbols denote the configuration of the final steady states, the final states marked "e"–"h" are those shown in the columns indicated of Figure 3, and the two horizontal lines marked "Threshold" indicate the mass that divides transient states which regrow to nearly the initial steady-state mass (EWNS or EWN configurations) from those which regrow only partially ("no-north" configurations: EWS, EW and E).

### 3.3 Interaction of ice-sheet and climate during decline

To demonstrate the important influence of the climate–ice-sheet interaction, we repeat the HadGEM2-ES abrupt4xCO2 low-
albedo experiment (the case of most negative ΔSMB) using Glimmer alone, uncoupled from the AGCM, forced by the AGCM
SMB field (a function of geographical location and tile elevation) from the start of the FiG experiment. As the uncoupled ex-
periment runs, the time-independent three-dimensional AGCM SMB field is continually interpolated onto the time-dependent
ice topography using the same methods as in the FiG coupling. Thus the local lapse-rate feedback on SMB is included in
the uncoupled experiment, but the regional climate feedbacks of topography and albedo change on the atmospheric state and
circulation are excluded.

The uncoupled Glimmer and FiG experiments begin from the same initial state and have the same initial rate of mass-loss,
but soon diverge (the dotted red line and the lowest solid red line in Figure 4a). While the rate of mass-loss continuously
decreases in the FiG experiment, it remains almost constant (2.1–2.6 mm yr$^{-1}$ SLE) in the uncoupled experiment for about
2.5 kyr, and the ice-sheet is completely eliminated in 3.4 kyr (Figure 4b).

To understand the different behaviour, as an example we compare the state when $M = 2.38$ m SLE, which is reached
after 3600 years in the coupled experiment and 2020 years in the uncoupled. The coupled ice-sheet has a high central region
(Figure 3c1), where specific SMB exceeds 0.25 m yr$^{-1}$ LWE over about the same area as in the initial state (Figure 3b2,c2),
surrounded closely by steep narrow margins with large negative specific SMB, giving negative area-integral SMB which is ~3
times smaller in magnitude than in the initial state under 4xCO2. The regions where negative specific SMB appears were near
to equilibrium in the initial state, and the change is consistent with the local lapse-rate feedback due to the lowered surface in
the contracted margins. The area-mean ratio of changes in surface air temperature and surface elevation is 7.1 and 6.6 K km$^{-1}$
within the initial and contracted ice-sheet extent respectively, close to the value of 6 K km$^{-1}$ assumed in the downscaling
scheme. It is not uniform over the ice-sheet (Figure S3), but it is within in the range 4–8 K km$^{-1}$ over more than half of the
ice-sheet (considering either extent).

The uncoupled ice-sheet is similarly located in the north of Greenland, but has larger area and lower altitude (Figure 3d1).
Its specific SMB is negative *everywhere*. Its area-integral SMB ($-991$ Gt yr$^{-1}$, Figure 3d2) is *more* negative than in the initial
state ($-756$ Gt yr$^{-1}$, Figure 3b2), and ~4 times more than for the coupled ice-sheet of the same $M$ ($-262$ Gt yr$^{-1}$, Figure 3c2).
The much larger change exceeds the lapse-rate effect, and the area-mean specific SMB for any surface altitude above 1000 m
is more negative in the uncoupled case than the coupled. The main cause is greater downwelling shortwave radiation at the
surface in the uncoupled case (Figure 3c3,d3), due to lower cloud fraction. The region occupied by the contracted ice-sheet
coincides geographically with the high cold interior of the initial ice-sheet, where cloudiness is comparatively low, but in the
coupled case the cloudiness increases there as the ice-sheet becomes smaller and lower, giving a powerful negative feedback
on the mass-loss.

In the coupled experiment, the precipitation from the south-west advances inland, following the the margin of the contracting
ice-sheet (compare the grey contour line for 1 m yr$^{-1}$ in Figure 3b4,c4). Consequently the precipitation on the ice-sheet is
about 15% greater in the coupled case. However, the snowfall is about 15% *less* in the coupled case (colours and numbers

in Figure 3c4,d4), because its surface is lower than in the initial climate, making the surface climate warmer and reducing the snowfall fraction (to 64%). The uncoupled SMB has a larger snowfall fraction (84%) because the surface in the region it occupies was initially much higher. The phase change of precipitation with elevation is omitted from the downscaling in the coupling scheme (as mentioned earlier); including it in the uncoupled model would reduce the snowfall, and make its SMB even more negative.

In summary, the uncoupled ice-sheet is eliminated rapidly through the small ice-cap instability (local lapse-rate feedbacks from surface energy fluxes and temperature), whereas in the coupled case the decline is decelerated, and the ice-sheet not completely eliminated, owing to negative feedbacks of topographic change on regional climate (changes in cloudiness and precipitation). The comparison demonstrates the critical role of ice-sheet–climate interaction.

## 3.4   Final topography of the ice-sheet

According to the topographic features present, the final states can be put in five categories (indicated by symbols at the ends of the trajectories in Figure 4b). In cases with small change in $M$, the final state is similar to the present day (configuration labelled "EWNS" *e.g.* Figure 3a1). The northern portion (denoted "N") is absent in some final states and the summit further south than in the present day *e.g.* Figure 3f1 (EWS). Ice in the south ("S") may become a separate ice-cap (as in Figure 3f1) or it may be absent, resembling Figure 3h1 (EWN) and 3g1 (EW). In cases with the smallest final $M$, the north-western lobe ("W") vanishes, and ice remains only on the eastern mountains ("E"). For example, in the experiment ending in Figure 3e1 (marked with "e" in Figure 4b), the southern and north-western domes detach and vanish within 3 kyr. Subsequently contraction continues on all sides, but there is a slow small regrowth after the minimum mass is reached.

The transient and final states of all experiments lie close to a common power-law relationship between ice-sheet mass $M$ and area $A$ with $M \propto A^{1.31}$ (Figure 2f), similar to the exponent of 1.36–1.38 derived for glaciers from observations and theory (Bahr and Radić, 2012, and references therein). Final states with the same configuration have a characteristic deviation from the common relationship *e.g.* EWN states have greater $M$.

Because of the local lapse-rate feedback, the mass-loss sometimes accelerates by a few tenths of a $\mathrm{mm\,yr^{-1}}$ SLE while one of the outlying portions becomes separate or is eliminated, in a few cases after some millennia of relatively slow change. This is a similar phenomenon to the saddle collapse during the separation of the Laurentide and Cordilleran ice-sheets during the last deglaciation (Gregoire et al., 2012), but an order of magnitude smaller. For example, in the experiment ending in Figure 3f1 (the dotted green line, marked "f", in Figure 4b), the rate of ice loss accelerates after 10 kyr, at the start of the retreat of the northern margin, which is completed by 15 kyr.

## 3.5   Discussion of reduced steady states

In Section 1.3 we described why there might be a threshold ΔSAT beyond which the ice-sheet would be eliminated by the small ice-cap instability, whereas with smaller ΔSAT it would have mass and area little reduced from its present-day state. In Section 3, instead of such a well-defined threshold, we found a range of steady-state ice-sheet mass and area, generally smaller for larger ΔSAT. The ice-sheet endures, albeit in a much reduced state, even for ΔSAT giving large negative initial SMB.

For studying the evolution of the ice-sheet as its area $A$ contracts, it is helpful to consider the specific SMB $s = S/A$, where obviously $S$ and $s$ have the same sign. We can write $\Delta s = \Delta s_T + \Delta s_L + \Delta s_C$, where $\Delta s_T$ and $\Delta s_L$ are the changes in specific SMB due to climate change and the local lapse-rate feedback, as in Section 1.3. When the warmer climate is initially imposed, $\Delta s_T < 0$, and the perturbation is amplified by $\Delta s_L < 0$ due to thinning of the ice-sheet.

The term $\Delta s_C$ represents the effects of change in the climate experienced by the ice-sheet, arising both because the climate changes in all areas, and because the ice-sheet changes the areas it occupies. An important example of the latter is the retreat of the ice-sheet margin (or a glacier tongue, in general) to higher altitude in a warmer climate, because this reduces the ablation while preserving the accumulation. In this and other cases, the climate effects can give $\Delta s_C > 0$. Thus they can counteract the local lapse-rate feedback $\Delta s_L < 0$, prevent a runaway feedback loop, and eventually reverse the sign of $\Delta s$ so that a steady state is reached with SMB and discharge in balance again, even if with greatly reduced area.

In cases where specific SMB is initially positive, it becomes more positive (Figure 2e), because the areas from which the ice-sheet retreats are predominantly those of relatively larger ablation or smaller snowfall. Consequently the area-integral SMB (the product of increasing specific SMB and decreasing area) changes relatively little (fairly vertical trajectories in Figure 2b). For instance, under MIROC5 RCP4.5 climate with medium albedo, the initial SMB, snowfall and ablation are 150, 735 and 585 Gt yr$^{-1}$ (Table 1). The final SMB is the same as the initial because ablation and snowfall both decrease by 115 Gt yr$^{-1}$ (Figure 2f2,f4), a larger fractional decline in ablation (20%) than in snowfall (15%). The steady state is achieved by the withdrawal of the margin from the coast in some sectors, reducing discharge sufficiently (by 209 Gt yr$^{-1}$ or 60%) to balance the smaller SMB.

In cases where specific and area-integral SMB are initially negative, they become positive (Figure 2b,e). This happens because snowfall decreases less than ablation. For instance, under HadGEM2 abrupt4xCO2 climate with low albedo, the initial SMB, snowfall and ablation are $-756$, 797 and 1554 Gt yr$^{-1}$ (Figure 2b2,b4). The final state is a small eastern ice-cap (like Figure 2e1 but smaller) with SMB, snowfall and ablation of 9, 31 and 21 Gt yr$^{-1}$; snowfall is 3.9% and ablation 1.4% of the initial value. The ice-cap receives greater precipitation and snowfall than the same region did initially (compare Figure 2b4,e4), and has more cloud and less surface downwelling shortwave (Figure 2b3,e3), because of the effect of topography on atmospheric circulation and climate.

## 4   Threshold for irreversible mass-loss

Greenland ice-sheet mass-loss in the first set of experiments occurs on timescales which are comparable with or even longer than those of surface climate change and natural $CO_2$ removal. We therefore also consider whether the ice-sheet mass would increase again if the climate cooled down. This will inform us about any irreversible commitment to GMSLR that might be incurred in coming decades despite subsequent $CO_2$ removal.

To study this question, we carry out a second set of FiG experiments, using MIROC5 1980-1999 BCs and recent radiative forcing, starting from various transient and final steady states of the ice-sheet with reduced size from the first set of experiments. This is as if the climate instantaneously reverted to its late 20th-century condition after many centuries in a high-$CO_2$ warm

steady state, during which the ice-sheet had been losing mass. The second set includes experiments with all three choices of albedo. All but one of the experiments with medium albedo (solid green lines in Figure 4c) begin from states of various mass

along the trajectory of the CanESM2 RCP8.5 medium-albedo experiment (green line with circles in Figure 4b), whose final steady-state ice-sheet mass is 1.21 m SLE. All but one of those with low albedo (solid red lines in Figure 4c) begin from states of the HadGEM2-ES abrupt4xCO2 low-albedo experiment (red line with circles in Figure 4b), whose final mass of 0.12 m SLE is the smallest of all in the first set. The single high-albedo experiment in the second set (solid blue line in Figure 4c) also begins from this minimal state. (The exceptions for medium and low albedo are the two experiments discussed in Section 4.2

and shown with dashed lines in Figure 4c, which begin from the final states of the experiments shown with dashed lines in Figure 4b.)

## 4.1 Regrown steady states

In the initial state of all the experiments of the second set, the ice-sheet has a smaller mass than present, and it grows to reach a new steady state; there are none in which it continues to lose mass (Figure 4c). However, the mass of the regrown steady state

depends on the initial state and the albedo.

 With high albedo, the ice-sheet regrows, in about 50 kyr, from the minimal state to a steady state with the extent of the present day's (EWNS configuration, Figure 4c). Since this starting state is a limiting case, we assume that the ice-sheet would reach the same final state from any initial state, implying that this is only steady state for historical climate with high albedo. Therefore the loss of the ice-sheet would be reversible, albeit on a long timescale, if the high albedo is realistic.

On the other hand, with the medium and low albedo, two distinct sets of steady states can be reached in the second set of experiments, one set with final mass of 7 m SLE or more, the other with final mass of 5–6 m SLE. Initial states are divided between these two sets of final states by a threshold of initial mass at 4.0 m SLE with the medium albedo, and 3.9 m SLE with the low albedo.

 Starting above the threshold, the ice-sheet regrows to the EWNS configuration with medium albedo as with high (Figure 4c),

but with low albedo there are two steady states. The larger is the EWNS configuration (7.3 m SLE, Figure 3a1), while the smaller lacks the southern dome (7.0 m SLE, EWN configuration, Figure 3h1). The southern dome has positive SMB in the historical climate (Figure 3a2), but in the warm climates it is readily lost due to increased ablation, and in the historical climate without the dome there is negative SMB inhibiting readvance at the new southern margin (Figure 3h2). The snowfall is however little changed in that region (Figures 3a4,h4). The southern dome is the last part of the ice-sheet to reappear with the high and

medium albedoes (solid green and blue lines in Figure 4c after 30 kyr).

 Starting below the threshold, the ice-sheet attains steady states lacking the northern portion, which we will refer to collectively as no-north states. The steady state with medium albedo has the EWS configuration (5.7 m SLE, like Figure 3f1 in extent but thicker). With the low albedo, there are two steady states, having masses of 5.3 m SLE (EW configuration, Figure 3g1) and 5.0 m SLE (E configuration, like Figure 3e1 but much larger), which differ because the north-western dome is missing in the

460 latter case. This dome is the last part to regrow with medium albedo.

Other authors have likewise found that the present state of the Greenland ice-sheet is not the only steady state under historical climate (Ridley et al., 2010; Solgaard and Langen, 2012; Robinson et al., 2012). A minimal state with ice solely or mostly in the east is a common feature of all these studies and ours. In other respects the steady-state configurations are dissimilar. The medium state of Robinson et al. (2012) most resembles our no-north states. Our results are more complex than others in showing five steady states. We suppose that this is because greater detail in the interaction of the ice-sheet topography with atmospheric circulation and SMB can be simulated by FAMOUS–ice than by the simpler approaches of previous studies.

## 4.2 Vulnerability of the ice-sheet to irreversible loss

To summarise our second set of experiments: transient states which have passed below the threshold regrow to no-north steady states, while those still above the threshold regrow to EWNS or EWN steady states. Consistent with this, we note that all final states lying below the threshold in the first set of experiments are no-north states (Figure 4b). States taken from below the threshold on trajectories of rapid decline show no tendency for the northern portion to regrow, even after tens of millennia under historical climate. Thus about 2 m of GMSLR will become irreversible once the Greenland ice-sheet mass drops below the threshold, if the medium or low albedo is realistic.

Under the same BCs, initial states which differ only slightly in topography (the minimum separation of our initial states in $M$ is actually 0.2 m SLE) can lead to final states which differ substantially (by more than 1 m SLE) because ice-sheet–climate feedbacks amplify the initially small difference in SMB. The probable reason is that ablation exceeds accumulation in the northern region without the ice-sheet (shown by negative SMB at the northern margin in Figure 3g2), partly because snowfall is reduced (Figure 3a4,g4).

The low- and medium-albedo no-north steady states following regrowth are 1–2 m SLE above the threshold, and yet grow no further, unlike states along trajectories of rapid decline having $M$ in the same mass range *i.e.* between the threshold mass (4 m SLE) and the no-north mass (5–6 m SLE). The implication is that, for states in this mass range, the outcome depends on the history. To test this, we have conducted further experiments (dashed lines in Figure 4c) beginning from the two steady states in this range (large circles at the end of dashed lines in Figure 4b), which were reached by *slowly* declining trajectories. These two are no-north (EWS) states. Initially the ice-sheet mass grows but, unlike when starting from rapidly declining transient states in this range, it soon becomes nearly constant at a slightly higher $M$ than is reached from states below the threshold. The difference in $M$ is due to a large southern dome, which was kept during the slow decline (along the dashed lines leading to the large circles in Figure 4b) but had been lost already in states of the same mass in the warmer climate producing the fast decline (the solid lines with red and green circles in Figure 4b), and is not rebuilt in the historical climate. This result suggests that, for slow or quasi-static decline of the ice-sheet, the no-north mass itself is the threshold of irreversibility.

Using the linear relationship between the initial rate of mass-loss and the final steady-state mass in the first set of experiments (solid line in Figure 2d), we can translate the threshold of irreversibility ($M$=3.9–4.0 m, horizontal red and green lines), which applies during trajectories of rapid decline, into a threshold on the *rate* of loss (vertical red and green lines). Under a warm climate which initially gives a more negative ΔSMB than the threshold rate, the ice-sheet will eventually decline to a state which is smaller than the threshold mass. Roughly estimating a range from the scatter in the relationship, the results suggest

that the threshold $\Delta$SMB lies between $-500$ and $-150$ Gt yr$^{-1}$ (Figure 2d). Since recently observed $\Delta$SMB (*e.g.* van den Broeke et al., 2016) is at the upper end of this range *i.e.* a relatively small rate of decline, we recall from the previous paragraph that the relevant threshold may instead be the no-north steady-state mass of about 5.5 m SLE. In that case the linear fit indicates that the recently observed $\Delta$SMB is close to the threshold rate which will eventually lead to partially irreversible loss of the ice-sheet.

If the recently observed rate of mass-loss of about 0.7 mm yr$^{-1}$ SLE persisted, it would take 4900 years for the ice-sheet mass to reach the threshold of irreversibility, and about 2700 years to reach the no-north steady-state mass. At the highest rate of loss simulated in our experiments for the end of this century, of about 2 mm yr$^{-1}$ SLE, it would take 1700 years to reach the threshold. Allowing for systematic uncertainty, the AR5 predicted even larger rates of mass-loss due to SMB perturbation, of up to about 6 mm yr$^{-1}$ SLE by the end of the century, at which rate the threshold would be reached in 600 years.

In order to avoid eventual irreversible ice-loss, the climate must be returned to near pre-industrial before the threshold mass is reached. Reversing climate change requires extracting heat from the ocean, as well as removing the radiative forcing. If that can be done at all, it could not be done instantaneously, and mitigating climate change in the short term will buy more time to save the ice-sheet on the long term. Further simulations would be required to evaluate whether particular trajectories of future climate would avoid irreversible ice-loss.

## 5 Conclusions

We have studied the multimillennial future evolution of the Greenland ice-sheet for various magnitudes of anthropogenic climate change, in experiments with constant climates using an AGCM interactively coupled to a dynamic ice-sheet model. For adequate resolution of gradients, especially at the margins of the ice-sheet, the surface mass balance is simulated by the AGCM as a function of elevation within its gridboxes. Our aim is not to produce time-dependent projections for coming centuries, but instead to investigate the long-term consequences for global-mean sea-level rise (GMSLR).

Under constant climates that are warmer than the late 20th century, the ice-sheet loses mass, its surface elevation decreases and its surface climate becomes warmer. This gives a positive feedback on mass-loss, but it is outweighed by the negative feedbacks due to declining ablation area and increasing cloudiness over the interior as the ice-sheet contracts. In the ice-sheet area-integral, snowfall decreases less than ablation because the precipitation on the margins is enhanced by the topographic gradient, and moves inland as the ice-sheet retreats. Consequently after many millennia under a constant warm climate the ice-sheet reaches a reduced steady state. Final GMSLR is less than 1.5 m in most late-21st-century RCP2.6 climates, and more than 4 m in all late-21st-century RCP8.5 climates. For warming exceeding 3 K, the ice-sheet would be mostly lost, and its contribution to GMSLR would exceed 5 m.

Contrary to expectation based on work using simpler climate models (Huybrechts et al., 1991; Gregory et al., 2004; Gregory and Huybrechts, 2006; Robinson et al., 2012; van den Broeke et al., 2016; Pattyn et al., 2018), we do not find a sharp threshold in regional Greenland or global warming that divides scenarios in which the ice-sheet suffers little reduction in its final steady state from those which it is mostly lost. Our results give a qualitatively different impression, because the transition occurs

over a larger temperature interval, or involves a smaller mass-loss. We think that this difference arises from our using an AGCM, whose dynamics and physical detail are needed to simulate the response of snowfall and cloudiness to the evolving topography. Support for this hypothesis comes from comparison with an experiment using the uncoupled ice-sheet model, in which the surface mass balance evolves only through the local lapse-rate feedback, and regional climate feedbacks are omitted. In that case an almost constant rate of mass-loss is maintained for 3 kyr, during which the ice-sheet vanishes completely.

Under a warm climate, the final ice-sheet mass, and the entailed commitment to GMSLR, are well-correlated with the initial perturbation to surface mass balance, and hence with the magnitude of climate change imposed. The final mass is also affected by the geographical pattern of climate change. According to a linear regression of our results, if a climate giving an SMB similar to that recently observed were maintained indefinitely, Greenland ice-sheet mass-loss would produce 0.5–2.5 m of GMSLR.

When transient and steady states of the ice-sheet obtained under warm climates are transplanted into the late 20th-century climate, as if subsequent anthropogenic climate change had been reversed, the ice-sheet regrows in all cases, over tens of millennia, but not necessarily to its present-day size (as also found by Charbit et al., 2008; Ridley et al., 2010; Robinson et al., 2012). The resulting steady states can be put in two groups, according to whether ice is present in the northern part of the island. If the ice-sheet retreats from this region, it may not regrow, because the snowfall is reduced there, meaning that about 2 m of GMSLR would become irreversible. This threshold size might eventually be reached with late 20th-century climate, and would be reached in about 600 years with the greatest rates of mass-loss projected for 2100 under RCP8.5 by Church et al. (2013). In order to avoid irreversible GMSLR, it would be necessary to restore the late 20th-century climate, in which the ice-sheet was near to mass balance, before the threshold is crossed.

The reliability of our conclusions depends on the realism of our model. There are systematic uncertainties arising from assumptions made in its formulation. The atmosphere GCM has low resolution and comparatively simple parametrisation schemes. The ice-sheet model does not simulate rapid ice-sheet dynamics; this certainly means that it underestimates the rate of ice-sheet mass-loss in coming decades, but we do not know what effect this has on the eventual steady states, which are our focus. The SMB scheme uses a uniform air temperature lapse rate and omits the phase change of precipitation in the downscaling from GCM to ice-sheet model. The snow albedo is a particularly important uncertainty; with our highest choice of albedo, removal of the ice-sheet is reversible.

Notwithstanding these limitations, our results demonstrate the importance of climate–ice-sheet interaction to projecting the future of the Greenland ice-sheet. It would obviously be useful if similar investigations were done using other models that couple an ice-sheet to an atmosphere GCM (perhaps as components of an AOGCM or Earth system model), especially with higher resolution in both the atmosphere and ice-sheet components. Even with our low-resolution GCM, large ensembles of long experiments are computationally demanding, and our results give only an outline of possible behaviour. They could be supplemented by using an emulator to explore a wider range of scenarios (Edwards et al., 2019).

*Data availability.* The model data used in this analysis will be deposited and made freely available for research at the Centre for Environmental Data Analysis upon publication of this paper. See www.met.rdg.ac.uk/~jonathan/data/gregory20greenland for further information.

## Appendix A:  Technical sensitivity tests of FAMOUS–ice–Glimmer

In order to test the sensitivity to certain technical changes, we ran three modified versions of the FAMOUS–ice–Glimmer experiment with CanESM2 RCP8.5 climate and medium albedo (from which the medium-albedo experiments of Section 4.1
begin, shown by a green line with circles in Figure 4b, and the solid black line in Figure S1a). The ice-sheet mass in each of the modified experiments differs by less than 0.2 m SLE from the standard experiment during the first 2000 years (Figure S1a).

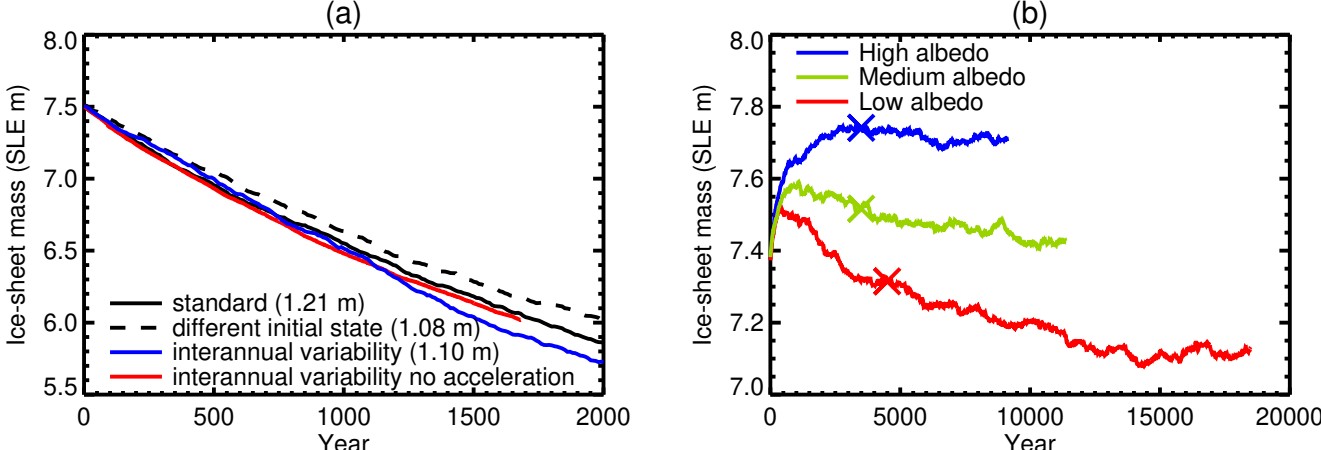

**Figure S1.** (a) Timeseries of Greenland ice-sheet mass for the first 2000 ice-sheet years in experiments with CanESM2 RCP8.5 2081–2100 climate and FAMOUS–ice medium albedo as an example of sensitivity to technical modifications: different AGCM initial state, individual monthly means for the sea-surface BCs ("interannual variabilty") rather than climatological monthly means, and synchronous coupling (one ice-sheet year per climate year, "no acceleration") rather than 10:1 acceleration. The numbers in parentheses give the final steady-state mass. (b) Timeseries of Greenland ice-sheet mass with constant climate for 1980–1999 simulated by MIROC5 during FiG spinup integrations beginning from the observed topography (Bamber et al., 2001a, b). The crosses indicate the states from which the experiments of Section 3 were initiated.

The ice-mass $M(t)$ in the first modified experiment (dashed black line in Figure S1a) remains within $\pm 0.2$ m of the standard experiment throughout its length, and is 0.1 m less than the standard in the final steady state. It is identical in forcing to the standard experiment but begins from a different atmosphere initial state of the same historical climate. Therefore its deviation
from the standard experiment is due to chaotic unforced climate variability alone. The size of this unforced deviation is small compared with the differences of outcome due to climate and albedo among the experiments discussed in the paper, showing that the forced differences are statistically significant.

For its sea-surface BCs, the second experiment cycles repeatedly through the 20-year series of individual monthly means that were used to make the 20-year climatological monthly means of the standard experiment. Thus it contains interannual variability in the climate. Its $M(t)$ (blue line in Figure S1a) is always within $\pm 0.4$ m of and in the end 0.1 m less than the standard experiment's.

The third experiment has the same BCs as the second, and differs in addition from the standard experiment in that the ice-sheet model is run for only one year (not ten) after each FAMOUS–ice year. Because this version is almost ten times slower, we ran it for only 1700 years. During that period, it differed in $M(t)$ by less than 0.05 m from the standard experiment (red line in Figure S1a). The second and third experiments both have different SMB in every FAMOUS–ice year, but the acceleration (in the second experiment) makes these persist for a decade in the ice-sheet model. We think this explains the greater excursions of the second experiment from the standard model.

## Appendix B: Relationship of albedo to steady-state historical ice-sheet mass

Continuing the spinup experiments (which are among the experiments of Section 3), ice-sheet mass $M$ remains at 7.7 m SLE with low albedo, with medium albedo it declines slightly to a steady state of 7.4 m SLE (very close to observed) over about 10 kyr, and with low albedo to 7.1 m SLE over 15 kyr (Figure S1b). These are small changes compared with those simulated for 21st-century climate change (Section 3). Nonetheless, these small differences in $M$ for low and high albedo from observations show that requiring a a realistic steady state of the ice-sheet in a coupled model provides a strong constraint on the SMB simulation to which regional climate models such as MAR and RACMO are not subjected. A quadratic fit to the relationship between SMB and $M$ in FiG steady states with MIROC5 historical climate gives $M = 7.9$ m SLE for the SMB of 437 Gt yr$^{-1}$ simulated by MAR for this climate.

An even higher choice of albedo in FiG gave SMB of 610 Gt yr$^{-1}$ and a steady-state $M$ of 8.2 m SLE, and an even lower choice 195 Gt yr$^{-1}$ with $M$ tending towards a steady state substantially below 7.0 m. These values of SMB approximately bound the range of SMB variations in the 20th century reconstructed with MAR (Fettweis et al., 2017, their Figure 8a), indicating that they could plausibly occur with historical climate and the present-day ice-sheet topography (as in MAR), but we excluded those choices of albedo because they would not be consistent with realistic $M$.

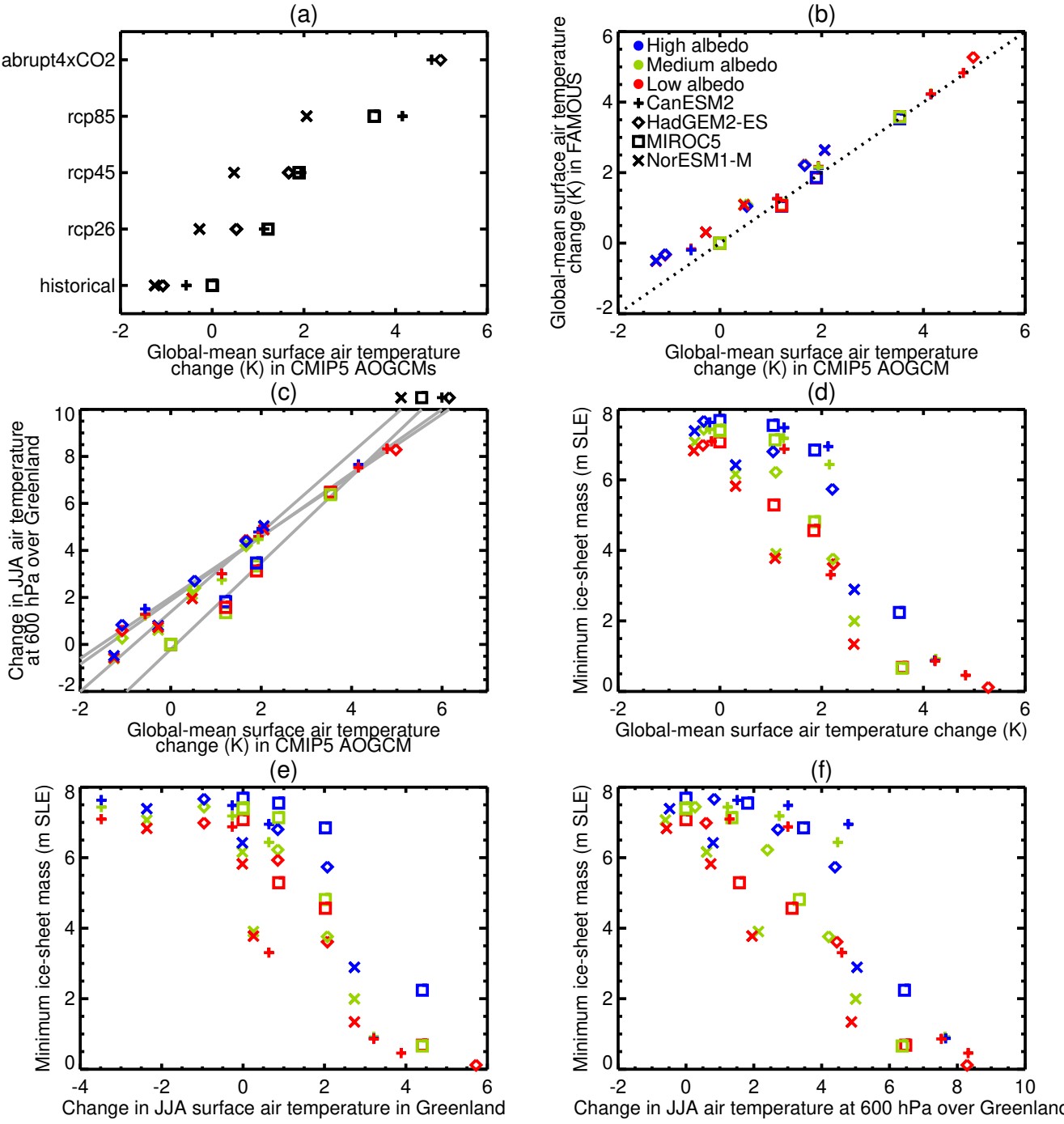

**Figure S2.** Relationships among air temperature change and Greenland SMB change in CMIP5 AOGCM and FAMOUS–ice–Greenland experiments. The dotted line in (b) is 1:1. The grey lines in (c) are regression lines for the subset of data from each of the four AOGCMs, indicated by the symbols along the top edge. (d) is the same as Figure 2c and repeated here for comparison. Changes are computed from the first 300 ice-sheet years and expressed relative to the MIROC5 historical climate with the same albedo parameter.

## Appendix C:  Alternative measures of air temperature change

In Figure 2 we obtain global-mean annual-mean surface air temperature (SAT) change, denoted ΔSAT, from FAMOUS–ice. It may also be obtained from the AOGCMs that supply the sea-surface BCs (Figure S2a). Separating the values by scenario reveals the AOGCM-dependence of global-mean SAT. NorESM1-M is cooler in general. HadGEM2-ES and CanESM2 have higher climate sensitivity, meaning that they warm more in response to forcing, while NorESM1-M has lower climate sensitivity and warms less.

ΔSAT in the CMIP5 AOGCMs is almost the same as in FAMOUS–ice (Figure S2b). They differ because SAT is not pre-scribed over land from the BCs in FAMOUS–ice.

We have investigated two other measures of air temperature change but neither is a better predictor than ΔSAT for the final ice-sheet mass (Figure S2e,f).

## Appendix D:  Lapse rate

In the downscaling of surface air temperature from FAMOUS gridboxes to FAMOUS–ice elevation tiles, we assume a uniform lapse rate of $6 \, \mathrm{K \, km^{-1}}$. Consequently this lapse rate is also used to predict the derivative of surface air temperature with respect to elevation change when Glimmer is run uncoupled from the AGCM. The derivative diagnosed from the coupled experiment is shown in Figure S3.

*Author contributions.*  SEG and RSS developed the model, SEG and JMG ran the experiments, JMG carried out the analysis and wrote the paper.

*Competing interests.*  The authors have no competing interests.

*Acknowledgements.*  We are grateful to John Church and Jason Lowe for often emphasising the importance of the long-term future of Greenland and GMSLR and thus motivating the research. We thank the reviewers (Alex Robinson, Xavier Fettweis and one other) for their thoughtful and constructive comments, which helped us to improve the paper. This work was supported by NERC grants NE/P014976/1 and NE/I011099/1.

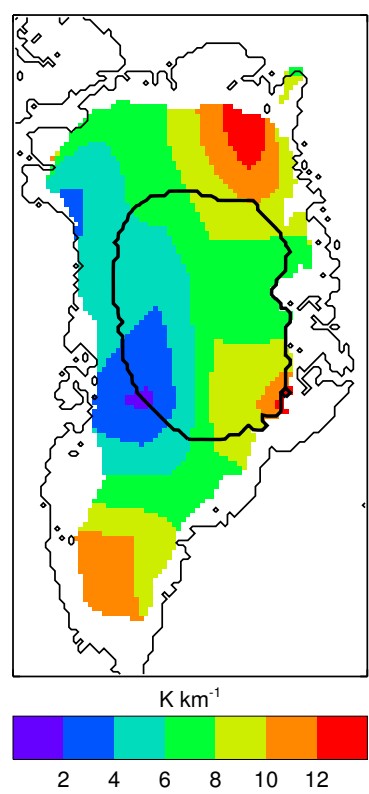

**Figure S3.** Change in surface air temperature divided by change in surface altitude ($K\,km^{-1}$) in the difference between the initial state of the experiment with HadGEM2-ES abrupt4xCO2 climate and low albedo and the state after 3600 years (Figures 3b1,c1). The thick black line is the ice-sheet edge in the latter state.

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
