# Peer review of "Large and irreversible future decline of the Greenland ice-sheet"

_The Cryosphere, 2020_

## Referee Comment (RC1) · Anonymous Referee #1 · 18 Jul 2020

Overall impression

This paper was much needed. The topic is very relevant and here it is addressed with a novel approach where a relatively course resolution Atmosphere General Circulation Model (AGCM) is forced with ocean conditions from several Atmosphere-Ocean General Circulation Model, and bidirectionally coupled with an ice sheet model. This permits multiple simulations with models of diverse climate sensitivity and diverse imprints of climate change on the Arctic/Greenland region.

The study provides novel insights into ice-climate feedbacks and the future long-term deglaciation of Greenland.

General comments

[Figure]

The authors present a study on the long-term stability of the Greenland ice sheet, a topic that has been addressed before, but in this case with a novel methodology, with novel conclusions inferred from the simulations, and empirical relationships obtained from these simulations. They use a course-resolution (7.5 degrees lon x 5 degrees lat) atmospheric GCM including SMB simulation on "tiles" with ocean forcing from four different climate models and three choices of albedo. They perform a total of 47 simulations, with 46 coupled to an ice sheet model (shallow-ice approximation, 20 km resolution) and 1 without coupling (no feedbacks from ice sheet change). I find the methodology of this study very efficient in addressing a complex problem with a sophisticated tool (coupled climate, SMB and ice sheet models including SMB downscaling via "tiles") but with adequate simplifications that solve the computational problem.

The paper is well written, with a detailed introduction on the problem of ice sheet stability. In the current form, the theory and conclusions are relatively easy to follow; however I find difficult to navigate the results, e.g., figures (with much dependencies and going forth and back in the text) or trace individually the outcome of the 47 simulations. To this point, I have included some comments regarding figure legends and keys, but if the readers can find further ways to help paper navigation, that would be helpful

I have three major comments/points of discussion regarding the main conclusions of the study:

The authors found no evidence for "warming threshold beyond which the ice sheet would be eliminated". However, one could argue that this depends on how this "threshold" and "eliminated" are defined. The authors based the claim on that "the ice sheet endures, albeit in a much reduced state", others could call this elimination. They use also use the claim that there is a large spread on the final ice sheet mass for the same global temperature change, coming from the albedo and boundary (ocean) conditions. However, if I look at figure 2c, I could draw two perpendicular lines intersecting at (2.6 K, 3 m SLE) and claim that all simulations with warming of more than ~2.6 K would result in very small ice sheets of less than 3 m sea level equivalent, and that no simulation with less than this warming results in final states of less than 3 m sea level equivalent. Yes, for 2 K of warming the spread in the final volume is very large (from almost no change to more than $\frac{1}{2}$ of the original size), but the spread does not include very small ice sheets. Beyond the 2.6 K "threshold" the spread does not include high volume final ice sheets.

The authors label the negative feedbacks found (cloud, precipitation) as "area feedbacks", as opposed to positive "thickness feedbacks" (elevation), but I am not very clear about this separation. For instance, the present-day distribution of solar radiation and precipitation is related not only to the margin position but to the surface topography (or thickness, Ettema et al, 2010; also see Figure 3c3,d3 and Figure 4c4,d4 where solar radiation and snowfall change within the common area as well). Likewise, the albedo feedback is a positive feedback and the albedo effect from area retreat may be included in its definition (besides the melt effects over the remaining ice sheet).

Finally, some of the conclusions of the study come from a empirical relationships inferred from this study's simulations, and not directly from the simulations. The distinction from inferred conclusions and direct conclusions should be made. For instance, when these relationships are applied to present-day rates (as there are actual simulations under present-day climate, but these are "stable" spin-ups) or to results from AR5 (e.g, projections for RCP8.5)

Technical comments/questions

The use of a surface energy balance calculation for melt (as opposed to empirical calculations, e.g., PDDs) should be highlighted more, for instance in the abstract. More detail on such calculation (e.g., energy fluxes, snowpack model, refreezing calculation, albedo calculation) should be given here, instead of only referring to a paper in preparation.

Very little attention is given to refreezing in the manuscript, how is refreezing evolving within the simulations?

Which kind of vegetation cover is simulated under the retreating ice sheet? How different are the properties of this land cover with respect to a glaciated surface?

I find a bit confusing that the spinup is made under 1980-1999 MIROC (ocean) boundary conditions, but scenario climates from other three models are applied without going through the historical period of the corresponding model. For instance, in Figure2a, the change in SMB is referred to historical MIROC, but in the text the historical value for a different model is given for this change in SMB. In Figure 3 historical HadCM3 SMB is depicted in column a, while the text in section 6 indicates that this climate is not used.

Specific comments

Abstract

The abstract is difficult to understand without having read the body text first. Much context is missing on the numerous complex statements. I suggest to give more context and/or reduce/generalize the conclusions.

Line7 – clarify meaning of "initially", and of "warming". A bit of introduction on the simulation design (e.g., "steady state", use of BC from four different models) could be helpful. Line 8 – "for all RCP8.5 climates" – clarify that this means for a steady state climate corresponding to 2081-2100, and not RCP/ECP8.5 up to 2300 as e.g, in Aschwanden et al, 2019, Vizcaino et al, 2015. Line 9 – "if recent climate were maintained" : this is a conclusion not from the simulations, but from empirical relationships obtained from the simulations. Line 11 – "the dominant effect is reduction of area" : effect on what? The statement is very cryptic, why is the area important? Line 12 – "the geographical variation of SMB must be taken into account". This seems to imply that this is not taken into account in previous work, but one could argue that simulation of ablation and accumulation area is a geographical variation already... Please specify further "geographical variation of SMB" Line 14 – "if late twentieth-century climate is restored (. . .) the ice sheet will not regrow to its present extent" – this would be straightforward after reading line 9, but this line 14 is based directly on the model simulations, and not

on the empirical relationships obtained... the mixing of direct and indirect results is a bit confusing. Line 15 "owing to such effects": which effects? The cloud and snowfall effects were just said to increase the SMB, so they would help to re-grow the ice sheet?

Section 1

L71 "Because of the elevation and albedo feedbacks (...) the present ice-sheet could not be regenerated" – Is a decrease of albedo "irreversible"?

Section 3

Famous-ice AGCM: can you explain how refreezing is calculated? Text in general lacks discussion of effects of refreezing (both in introduction and discussion)

L123-124 Muntjwerf et al., GRL, 2020 also use this approach with same number of elevation classes for CESM2.0 L126 Sellevold et al, TC, 2019 discusses the sensitivity of the "tiles" downscaling method to lapse rate choice L129-L120: which gradients are these, can you specify? L134: "There is an uncertain parameter (...)" please specify L137: since Smith et al is in preparation, could you give here more detail on the albedo modeling? L151-L157: how is sliding parameterized in the ice sheet model? L163: "we run 10 years", do you mean, each atmospheric year is used 10 times, then next atmospheric year is run, etc,... L165 -> "we have verified" : would it be possible to show some proof of this, e.g., a figure in the supplementary information? Initialization L176 Title "3.3" can be more precise, maybe specify "(Evaluation of) Simulated 1980-1999 surface mass balance"? L193: "similar" is perhaps subjective. Please discuss similarities/differences Table 1, last sentence is unclear (20, 30 or 100 years, "second group"?) The SMB for RCP2.6 is very similar to historical. This makes sense as only the 2080-2099 (steady-state) climate is used, as compared to other SMB estimates with evolving ice sheet topography where the full 21st century climate for RCP2.6 is applied. Maybe good to add some cautionary text to avoid misleading comparisons?

Table 2 Why 402 ppmv for the historical period, isn't it too high? The mean concentration was lower than that in 1980-1999.

Section 4. L215 In the legend of Figure 2 it says delta_SMB is referred to steady state under historical MIROC5, but here the HadGEM2-ES value (+307) is given, which one is correct?

L220 − 10% larger -> I get 14%, am I missing something? (0.67*1.50/0.88) L238 "becomes gradually more positive" -> increases

L269 "there are such states" : Could you add detail on those? It is difficult to map them from Figure 2d to Figure2b to follow the SMB evolution.

L292 smaller -> lower L299 "more negative" "4 times more" – confusing, please give values L307-308 precipitation contours are difficult to read or labels are absent

Section 6 L368, L371 "All but one" : are you explaining this "one" somewhere else? L398 "we suppose this dome might regrow in time": it seems it does not regrow in 20,000 years, when do you expect it will start to regrow? L411 "difference" in what? "infinitesimal" -> what does this mean? Please quantify L435-439 Why 2,500 years at 0.7 mm/yr and 1,700 at more than double the rate (2 mm/yr)? How have you done these calculations? Which GrIS mass are you considering as present-day mass and as NON steady-state mass? L452 "outweighted" : in which sense? L453 "Snowfall": do you mean ice-sheet-integrated? L466 "If a climate (...)" There is a jump here. The results now are based on the empirical relationships, but not directly on the simulations here as the ice sheet is relatively stable under 1980-1999 forcing. L479-481 What about the course resolution of the atmospheric model, would you include it as a limitation of this study? L485 "sketchy" has a negative meaning in informal American English, maybe replace by other adjective ("gross")

Figures

Figure 1. White contour in b), is this modelled or observed "ice margin"? The ELA contour of c) makes a strange shape in the NW, any idea why?

Figure 2- It is difficult to read precipitation from the last row of figures. Only one contour line is labeled, there are very few others, and no line interval is given.

Figure 4 – Legend text and keys are confusing. My understanding is that the line color indicates albedo choice. "climates (…) indicated by the line colors according to the line key of (c)" would hint to the colors indicating climates, but it is actually albedo? I would take the line key from c) and put it in a common space as it applies to all panels. Same for the symbols in c) The orange dotted line corresponds to offline simulation with low albedo, shouldn't it be red dotted for consistency? Otherwise, add it to the key for lines, with a "Low albedo, offline run" Legend: "the circles indicate transient and final sites" Panel b- Legend text is not clear. Having circles in the key is confusing, as only the colors are used, and those correspond to the symbols in the key of c).

References

Aschwanden, A., Fahnestock, M. A., Truffer, M., Brinkerhoff, D. J., Hock, R., Khroulev, C., Mottram, R., & Khan, S. A. (2019). Contribution of the Greenland Ice Sheet to sea level over the next millennium. Science Advances, 5(6), eaav9396. https://doi.org/10.1126/sciadv.aav9396

Ettema, J., et al. (2010). "Climate of the Greenland ice sheet using a high-resolution climate model - Part 2: Near-surface climate and energy balance." Cryosphere 4(4): 529-544.

Sellevold, R., van Kampenhout, L., Lenaerts, J. T. M., Noël, B., Lipscomb, W. H., and Vizcaino, M.: Surface mass balance downscaling through elevation classes in an Earth system model: application to the Greenland ice sheet, The Cryosphere, 13, 3193–3208, https://doi.org/10.5194/tc-13-3193-2019, 2019.

---

## Referee Comment (RC2) · Alexander Robinson (Referee) · 31 Jul 2020

This study presents new simulations of the Greenland ice sheet under long timescale climate change to assess its stability. A new assessment of this nature is long overdue, since the widely cited threshold estimate is based on the results of only one study (Robinson et al., 2012). The work here is innovative primarily because the simulations are performed using an ice-sheet model that is fully (bidirectionally) coupled to a low resolution GCM climate model. The process-based analyses presented are a very valuable contribution to our understanding of this complex coupled system. Nonetheless, I believe significant revisions are needed to improve the clarity and precision of the study (in particular, because of the extra scrutiny and attention that this paper will likely receive in the context of AR6).

[Figure]

Major comments

Threshold or no threshold. The statements in this paper regarding the existence of a threshold for large-scale melting are framed as rather strongly, in a way that greatly contrasts with the results of Robinson et al. (2012) and previous work. However, I feel that this rather binary framing is not really warranted, nor does it help the community gain clarity on the issue.

- First, I would argue that the experimental setup here simply does not allow such a definitive conclusion to be made. Only 47 experiments are performed with rather arbitrary levels of SST warming applied based on available GCM experiments. This leaves some conclusions open to interpretation. For example, in Fig. 2c, it could be argued that the low-albedo model (red points) shows a roughly linear reduction in equilibrium volume as a function of temperature anomaly, while the high-albedo model (blue points) shows a threshold at ∼2 °C.

- Second, it is clear from this and previous work that strong, positive feedbacks do exist that give the system the potential for self-sustained melting of the ice sheet (albedo, elevation feedbacks). This work shows that there are additional important negative feedbacks (circulation changes leading to increased cloudiness and precipitation) that can serve to counterbalance the positive ones. Given this, I think the binary framing of "threshold or no threshold" is rather misleading.

- Third, this work will clearly feed into the upcoming IPCC report. Simply including the headline statement "There is no threshold" implies that these results run completely counter to previous work. But one could also make the following statement: "Above 2 °C, all simulations show retreat of the ice sheet to less than half of its current size." This statement is actually quite consistent with previous results, with the difference being how far the ice sheet retreats.

For the reasons above, I would suggest a general reframing of the discussion of these results in relation to previous work to highlight the continuity in our growing understanding of this complex system.

Section 2 ("Conceptual basis for the existence of a threshold warming"). Related to the point above, I don't quite see how this section adds value to the manuscript, as it is currently framed, especially since later it is stated that the "the conceptual basis for its existence is incorrect". It feels somewhat like a straw-man argument. The simple equations described in Section 2 are useful for conceptualizing the possibility of a runaway feedback leading to the complete melting of an ice sheet. But I think it is by now clear to the community that an ice sheet like Greenland is a large, complex system with processes coupled to atmospheric circulation and a wide range of acting timescales. It is clear, for example, that $\Delta s$, A and $f(\Delta M)$ will all change over time. Therefore, I would suggest to the authors that, rather than framing this as the current paradigm that should be rejected, it would be more valuable to highlight, conceptually, what could happen when some of those terms vary (i.e., when A becomes smaller, but $\Delta s$ increases, or to expand $f(\Delta M)$ into the multiple contributions that may exist, like f_{albedo}, which is most often positive in a warming climate, and f_{cloudiness}, which is found here to be an important negative feedback). Because I think the authors would agree that, ignoring possible climate feedbacks like cloudiness for a moment, as done in the uncoupled experiment, the theoretical basis for a threshold for ice sheet retreat (i.e., the small ice cap instability) still applies here. It is just mitigated by additional feedbacks/factors that are not accounted for in this simple equation.

Along the same lines, I don't think it makes sense to summarize the study of Robinson et al. (2012) at the end of this section as estimating $\Delta M$. In that study, as in this one, a fully coupled climate – ice-sheet system is simulated with a dynamically evolving ice sheet and topography. The results of such experiments allow later comparison with expectations from this conceptual framework, but this equation is not used at all for any quantitative analysis.

The scatterplot of Fig. 2c is indeed interesting. While above 2 °C, only rather low-volume states appear to be accessible, from $\sim$0.5-2.0 °C, a wide range of intermediate
states are accessible. It appears that the low- and medium-albedo model versions particularly allow access to volume distributions between 3-6 m sle. In contrast, the high-albedo model version mainly shows states with a large volume or a much smaller volume. Is there a reason that the high-albedo model may exhibit more threshold-like behavior than the low-albedo model? I think a discussion around this point would be a valuable addition to understanding the physics of the system.

The model setup is innovative and well-described. In particular, the use of elevation classes (i.e., tiles) is a proven method to improve the downscaling of smb to the higher resolution ice sheet model from low-resolution GCMs. Nonetheless, I have two key concerns:

1. I am very surprised to see an SIA model applied here, and at only 20km resolution. At a minimum, some justification of this relatively low grid resolution should be made (I would not expect computational cost to be an issue for such a model in this framework). More importantly, the authors should acknowledge and discuss the possible impact of a lack of fast ice dynamics in their simulations. For example, is basal sliding parameterized in some way, or is no basal sliding allowed? If it is not, the model is likely underestimating the dynamic adjustment of the ice sheet to the area retreat, which is an important positive feedback on ice decline on these timescales.

2. The FAMOUS atmosphere is necessarily low-resolution for computational speed, but 7.5° lon x 5.5° lat corresponds to roughly 7 grid points east-west and 5 grid points north-south over Greenland. Given that this study highlights the importance of atmospheric circulation changes impacting Greenland stability, such a low atmospheric resolution here seems problematic. Have the authors considered running a short experiment with a higher resolution equivalent of the AGCM with the same boundary forcing and a reduced ice-sheet configuration, but no active ice-sheet model, to see if the atmospheric state is similar to that predicted by the very low-resolution version? Such an experiment, if possible, would go very far towards understanding the possible uncertainties related to these non-linear feedbacks with the atmosphere.

It would also be valuable to see a figure showing the forcing applied to the model. For example, how do the present-day SST fields compare to reanalysis or observed SSTs? What are the future patterns of warming? Also, it generally seems that the simulations forced by NorESM1-M stand apart from the other two with lower ice volumes predicted for the same level of global warming. Is this reflected in the SST warming patterns in some way?

Finally, in terms of style, I find that the use of abbreviations for different variables throughout the text makes the manuscript harder to follow. For example, on L30, the phrase "The increase in D is probably the ice-dynamical response…" would be more straightforward replacing "D" with "discharge". Perhaps the authors could consider only using the variable abbreviations (P, D, R, M, etc.) when the text is related to specific equations that use them, and otherwise use the actual names in sentences. Some other abbreviations could be avoided all together (BCs, GMSLR, etc.).

Minor comments:

L11-12: "This is because the dominant effect is reduction of area, not reduction of surface altitude, and the geographical variation of SMB must be taken into account." <= This sentence could be more precise. In previous work geographical variation of SMB was also considered, even if in a simpler way. Nor does it seem that the dominant effect is the reduction in area. Rather, it seems that changes in atmospheric circulation act to mitigate the warming via increased cloudiness. The next sentence is already clearer, so I would suggest deleting this one.

L15: "owing to such effects," <= This reference is not very clear, as increased cloudiness and precipitation would, in principle, help the ice sheet regrow. Consider rephrasing, or simply removing.

~L118, Section 3.1: Other boundary conditions aside from topography and SSTs should be explicitly mentioned here. Are greenhouse gas concentrations applied in FAMOUS-ice to be consistent with the applied SST fields, for example?

Fig. 3: This is an important figure, but feels a bit busy. Perhaps the color bars of each row could be placed vertically on the side? This would clean it up a bit and make more room for the panels themselves.

Fig. 4: This figure is hard to follow, as there is a lot of information. I would suggest revising colors and symbols to provide more clarity. For example, I think the colors for different forcing scenarios should be substantially different from the colors delineating model versions (low, medium, high albedo).

L269-270: A stable but diminished ice sheet is consistent with previous work. Robinson et al. (2012) found that 10% of the ice sheet remained even above the tipping point for large scale melting (see state E3 shown in Fig. 4 of that reference for an approximate picture). It is clear that at some point retreat of the ice sheet to high elevation zones may lead to restabilization. I suggest reframing here.

L337-345: Consider reframing title of this section and first paragraph along the lines of earlier comments.

L387-388, L395, etc: I would remove the terms WOWS and NON, as they do not help comprehension beyond the already defined EWNS terms, and are used rather rarely in any case.

L411-412: I would point out that this seems to be an example of a tipping point being activated. This is, of course, not starting from present day, and so is not the same as what has been discussed until now. But it does show that the mechanisms for triggering self-sustained decline are present in the system.

L458: Greenland or global warming => regional or global warming

---

## Referee Comment (RC3) · Xavier Fettweis (Referee) · 9 Aug 2020

This paper presents important results about the irreversibility of the Greenland ice sheet based on a full coupled ice sheet-general circulation model. Although this paper is quite technical and is likely not accessible for a nonspecialist, it is well written, clear, innovative and certainly deserves to be published in TC. It notably confirms that the retreat of the ice sheet impacts more the future projections at the beginning of the simulations than the melt-elevation feedback, as shown by Le clec'h et al. (2019) using models at higher resolution.

However, in addition to the remarks of the 2 other reviewers, I have some additional minor remarks before acceptance of the paper.

[Figure]

1. In addition to Fig1, it should be interesting to show the differences/biases with the "reference". In the legend, is it MAR forced by MIROC5? or ERA?

2. Lines 183-187: The near-surface climate from RCMs forced by a GCM can not be compared directly with the near-surface climate from the forcing GCM as RCMs simulate their own boundary layer climate and are even able to correct near-surface GCM biases. As FAMOUS-ice is forced by near-surface climate from GCM, it is normal that there are differences with the RCMs simulations. The RCM are only sensitive to the free atmosphere climate from GCMs. Therefore, this section should be a bit rephrased to explain this issue.

3. Lines 314-317: one of the more important results of this study is the necessity of a full coupling with atmosphere to evaluate tipping point of the Greenland as changes in topography impacts on precipitation and cloudiness as negative feedback. I fully agree with this statement (as we have found the same when MAR has been coupled with GRISLI in Le clec'h et al.) but I have some reserve about the robustness of these simulated atmospheric changes in view of the spatial resolution (7.5°x 5°) used by the FAMOUS AOCGM. I understand the use of such a huge resolution in this study but an evaluation of these fields over current climate (by comparison with ERA-Interim) will be very useful to evaluate the ability of FAMOUS to simulate precipitation and cloudiness. At such resolution, how many pixels are there over the ice sheet and what is the topography over current climate. Finally, do changes in the Greenland topography impact on only the local climate over Greenland or the climate at a larger scale? This issue linked to the very low resolution used should at least be mentioned in the conclusion (Lines 478-486).

Le clec'h, S., Charbit, S., Quiquet, A., Fettweis, X., Dumas, C., Kageyama, M., Wyard, C., and Ritz, C.: Assessment of the Greenland ice sheet–atmosphere feedbacks for the next century with a regional atmospheric model coupled to an ice sheet model, The Cryosphere, 13, 373–395, https://doi.org/10.5194/tc-13-373-2019, 2019.

---

## Author Comment (AC1) · 18 Sep 2020

We are grateful to the anonymous referee, to Alex Robinson and Xavier Fettweis for their thoughtful and constructive comments. We hope that we have improved the paper by addressing their concerns, as outlined below, beginning with major points made by more than one referee.

Our revised manuscript with markup showing the changes is attached as a supplement to this comment.

**Threshold**

**Referee 1 (2)** *The authors found no evidence for "warming threshold beyond which the*

ice sheet would be eliminated". However, one could argue that this depends on how this "threshold" and "eliminated" are defined. The authors based the claim on that "the ice sheet endures, albeit in a much reduced state", others could call this elimination. They use also use the claim that there is a large spread on the final ice sheet mass for the same global temperature change, coming from the albedo and boundary (ocean) conditions. However, if I look at figure 2c, I could draw two perpendicular lines intersecting at (2.6 K, 3 m SLE) and claim that all simulations with warming of more than ~2.6 K would result in very small ice sheets of less than 3 m sea level equivalent, and that no simulation with less than this warming results in final states of less than 3 m sea level equivalent. Yes, for 2 K of warming the spread in the final volume is very large (from almost no change to more than 12 of the original size), but the spread does not include very small ice sheets. Beyond the 2.6 K "threshold" the spread does not include high volume final ice sheets.

**Referee 2 (1)** *Threshold or no threshold. The statements in this paper regarding the existence of a threshold for large-scale melting are framed as rather strongly, in a way that greatly contrasts with the results of Robinson* et al. *(2012) and previous work. However, I feel that this rather binary framing is not really warranted, nor does it help the community gain clarity on the issue.*

• *I would argue that the experimental setup here simply does not allow such a definitive conclusion to be made. Only 47 experiments are performed with rather arbitrary levels of SST warming applied based on available GCM experiments. This leaves some conclusions open to interpretation. For example, in Fig. 2c, it could be argued that the low-albedo model (red points) shows a roughly linear reduction in equilibrium volume as a function of temperature anomaly, while the high-albedo model (blue points) shows a threshold at* ~$2°$C.

• [A related later comment] *While above $2°$C only rather low-volume states appear to be accessible, from* ~$0.5$–$2.0°$C, *a wide range of intermediate states are accessible. It appears that the low- and medium-albedo model versions particularly allow access*

*to volume distributions between 3–6 m sle. In contrast, the high-albedo model version mainly shows states with a large volume or a much smaller volume. Is there a reason that the high-albedo model may exhibit more threshold-like behavior than the low-albedo model? I think a discussion around this point would be a valuable addition to understanding the physics of the system.*

*• It is clear from this and previous work that strong, positive feedbacks do exist that give the system the potential for self-sustained melting of the ice sheet (albedo, elevation feedbacks). This work shows that there are additional important negative feedbacks (circulation changes leading to increased cloudiness and precipitation) that can serve to counterbalance the positive ones. Given this, I think the binary framing of "threshold or no threshold" is rather misleading.*

*• This work will clearly feed into the upcoming IPCC report. Simply including the headline statement "There is no threshold" implies that these results run completely counter to previous work. But one could also make the following statement: "Above 2°C, all simulations show retreat of the ice sheet to less than half of its current size." This statement is actually quite consistent with previous results, with the difference being how far the ice sheet retreats.*

*For the reasons above, I would suggest a general reframing of the discussion of these results in relation to previous work to highlight the continuity in our growing understanding of this complex system.*

We appreciate the points that the referees are making. We agree on the need to be clear that our results are not revolutionary. We find that with small enough warming the ice-sheet is little reduced, while with large enough warming little of it remains, and those statements agree with previous results. However, the temperature interval between "small" and "large" is wider in our results, or alternatively the mass interval between "small" and "large" is narrower. Because of this, we think that our results give a qualitatively different impression from those of Levermann *et al.* (2013, using

the model of Robinson *et al.*)—see Figure 1 below. That is what we were intending to convey by the statement that, "There is no threshold." We think this is an important point to get across.

Levermann *et al.* describe their results as follows.

> The contribution to sea-level commitment from the Greenland Ice Sheet is relatively weak (on average 0.18 m °C$^{-1}$ up to 1°C and 0.34 m °C$^{-1}$ between 2 and 4°C) apart from the abrupt threshold of ice loss between 0.8 and 2.2°C above preindustrial (90% credible interval) (Fig. 1C). This corresponds to a transition from a fully ice-covered Greenland to an essentially ice-free state (*i.e.*, a reduction in ice volume to approximately 10% of the present-day volume, corresponding to a sea-level contribution of more than 6 m).

(Note: their text says "**of** approximately 10%"—we think they meant "**to** approximately 10%", or equivalently, "**by** approximately 90%".) It can be seen in Figure 1a that any given version of their model the temperature interval of the transition is very small—the lines look vertical. By comparison, for the low- and medium-albedo versions of our model, a similar transition between ice-sheet masses of about 6 m and 1 m SLE occurs over about a 2°C temperature interval. As referee 2 says, it might be that the high-albedo version of our model has a transition over less than 0.5°C between about 6 m and 3 m. More experiments are needed to probe that range. Even if so, that is less of a jump than in the model of Levermann *et al.*

We hope that the referees agree with the above assessment. Accordingly, we have removed the bald statement, "There is no threshold", and have revised the text of the conclusions, abstract and elsewhere describing the behaviour of our model and its differences from previously published ones. Both referees suggest a statement of the kind, "With warming exceeding $X$, the steady-state ice-sheet mass is smaller than $Y$,"

and we have included such a statement in the text and abstract. In view of the comment from referee 1, we now say "eliminated" only when we mean *completely* eliminated.

We have remarked on the need implied by referee 2 to explore the transition more thoroughly with further experiments, on the roles of both positive and negative feedbacks (see also response below concerning his point 2), and on the differences in behaviour between the versions of our model. We suggest that, if indeed the high-albedo version has a more abrupt threshold, it could be because the small ice-cap instability is intensified by positive albedo feedback, which is strongest in this version, but we have not included this speculation in the text.

**Low resolution**

**Referee 1, L479–481** *What about the coarse resolution of the atmospheric model, would you include it as a limitation of this study?*

**Referee 2 (2)** *The FAMOUS atmosphere is necessarily low-resolution for computational speed, but 7.5° lon × 5.5° lat corresponds to roughly 7 grid points east-west and 5 grid points north-south over Greenland. Given that this study highlights the importance of atmospheric circulation changes impacting Greenland stability, such a low atmospheric resolution here seems problematic. Have the authors considered running a short experiment with a higher resolution equivalent of the AGCM with the same boundary forcing and a reduced ice-sheet configuration, but no active ice-sheet model, to see if the atmospheric state is similar to that predicted by the very low-resolution version? Such an experiment, if possible, would go very far towards understanding the possible uncertainties related to these non-linear feedbacks with the atmosphere.*

**Referee 3, (3), Lines 314–317:** *One of the more important results of this study is the necessity of a full coupling with atmosphere to evaluate tipping point of the Greenland as changes in topography impacts on precipitation and cloudiness as negative feedback. I fully agree with this statement (as we have found the same when MAR has*

*been coupled with GRISLI in Le clec'h* et al.*) but I have some reserve about the robustness of these simulated atmospheric changes in view of the spatial resolution (7.5° × 5°) used by the FAMOUS AOCGM. I understand the use of such a huge resolution in this study but an evaluation of these fields over current climate (by comparison with ERA-Interim) will be very useful to evaluate the ability of FAMOUS to simulate precipitation and cloudiness. At such resolution, how many pixels are there over the ice sheet and what is the topography over current climate. Finally, do changes in the Greenland topography impact on only the local climate over Greenland or the climate at a larger scale? This issue linked to the very low resolution used should at least be mentioned in the conclusion (Lines 478–486).*

As all the referees suggest, we have now remarked in the conclusions that the low resolution of the GCM is a limitation of the model. As referee 2 says, Greenland spans only 7 by 5 grid-boxes in the free atmosphere of FAMOUS, and we have now remarked on this when the model is introduced. We have now also drawn attention to the comparison shown by Smith *et al.* (*Geosci. Model Develop. Discuss.*, https://doi.org/10.5194/gmd-2020-207) of the cumulative distributions of area as a function of altitude (the hypsometry) from FAMOUS and the subgridscale scheme. The SMB simulated with the subgridscale scheme has reasonably realistic features, as discussed in Section 3.3 (now 2.3), where we note that the precipitation gradients are probably too weak because of the low GCM resolution.

The evaluation of SMB and other quantities relevant to SMB for late 20th-century climate (as simulated by MIROC5) gives us some confidence in the simulations for climate change. Smith *et al.* present more information about this comparison. We have not carried out a simulation forced with ERA-interim because our experimental design uses surface BCs directly from the AOGCMs for the late 20th and 21st centuries (rather than, for instance, adding anomalies to observational BCs).

The SMB and AGCM–ice-sheet coupling schemes are technically complicated model developments, and it is not trivial to incorporate them in a different AGCM. Over the past

few years, at the same time as the work described in this paper and as part of a larger project involving a team of collaborators, we have implemented these schemes, including downscaling to 1.2 km for the Bisicles ice-dynamical model, in the present version of the UK Earth system model, whose atmosphere resolution is 1.875° longitude × 1.25° latitude (N96). (This model is computationally about 10,000 times more demanding than FAMOUS–ice.) Our climate-change experiments with UKESM–ice are at an early stage, and unfortunately it would be premature to include any results in the present paper. In a preliminary experiment of 1500 ice-sheet years under abrupt4xCO2 forcing, the Greenland ice-sheet contracts towards a shape like Figure 3c2 of the present paper, and the SMB increases in the central part of it. This is due at least partly to an increase in snowfall of a few $0.1\,\mathrm{m\,yr^{-1}}$ LWE because of the altered topography, by comparison (as the referee suggests) with another experiment in which the ice-sheet is not dynamic. We are unsure yet about the effects of cloud changes on SMB in this experiment. Because of the systematic uncertainty in any model investigation, it would clearly be valuable if similar experiments to ours were carried out with other coupled ice-sheet–climate models, as we say in the last paragraph of the paper.

The question of referee 3 about the remote climate effect of changes in Greenland is scientifically interesting. However, we feel it is not within the scope of this paper. We note that Ridley *et al.* (2005, 10.1175/JCLI3482.1) found the remote effects to be small, as simulated by HadCM3. We are grateful to Referee 3 for reminding us of Le clec'h *et al.* (2019), now cited, which is closely relevant to our work.

**Referee 1**

**(1)** *I find difficult to navigate the results, e.g., figures (with much dependencies and going forth and back in the text) or trace individually the outcome of the 47 simulations. To this point, I have included some comments regarding figure legends and keys, but if the readers can find further ways to help paper navigation, that would be helpful.*

We have followed the referee's suggestions, thank you. In revising the text we have

added some more cross-references as well, which may help.

**(2)** *See **Threshold**, above*

**(3)** *The authors label the negative feedbacks found (cloud, precipitation) as "area feed-backs", as opposed to positive "thickness feedbacks" (elevation), but I am not very clear about this separation. For instance, the present-day distribution of solar radiation and precipitation is related not only to the margin position but to the surface topography (or thickness, Ettema et al, 2010; also see Figure 3c3,d3 and Figure 4c4,d4 where solar radiation and snowfall change within the common area as well). Likewise, the albedo feedback is a positive feedback and the albedo effect from area retreat may be included in its definition (besides the melt effects over the remaining ice sheet).*

Yes, that is a good point. We have changed the text to refer to the elevation–SMB feedback included in the interpolation from FAMOUS to Glimmer (and present in the uncoupled model) as the "local lapse-rate feedbacks", and all the others (due to the response of the FAMOUS climate to the evolving ice-sheet) as "regional climate feed-backs".

**(4)** *Some of the conclusions . . . come from a empirical relationships inferred from this study's simulations, and not directly from the simulations. The distinction from inferred conclusions and direct conclusions should be made. For instance, when these relation-ships are applied to present-day rates (as there are actual simulations under present-day climate, but these are "stable" spin-ups) or to results from AR5 (e.g. projections for RCP8.5).*

We think that these concerns arise from our rather inconsistent use of "recent" and "present-day". In revising the text, we have been consistent in referring to 1980–1999 (whose simulated climate is used for spin-up to steady state) as "late 20th century", and observations of SMB and ice-sheet mass loss from the last couple of decades as "recent", while "present-day" is used only for the ice-sheet mass and topography, which changes comparatively slowly. The referee has made some specific comments about

this general point at line 466, to which we respond below.

**(5)** *The use of a surface energy balance calculation for melt (as opposed to empirical calculations,* e.g. *PDDs) should be highlighted more, for instance in the abstract. More detail on such calculation (*e.g. *energy fluxes, snowpack model, refreezing calculation, albedo calculation) should be given here, instead of only referring to a paper in preparation.*

We have inserted remarks comparing with empirical SMB schemes in Section 2 preamble and Section 2.1 and mentioned our approach in the abstract—thanks for the suggestions. We would prefer not to include more detail of the schemes in this paper. Please note that Smith *et al.* is presently under open review at https://doi.org/10.5194/gmd-2020-207 in *Geosci. Model Develop. Discuss.* It contains information on all the matters of interest, and the referee is welcome to comment if the information given there is insufficient.

**(6)** *Very little attention is given to refreezing in the manuscript. How is refreezing evolving within the simulations? Can you explain how refreezing is calculated? Text in general lacks discussion of effects of refreezing (both in introduction and discussion).*

Refreezing is described by Smith *et al.* (see previous comment).

**(7)** *Which kind of vegetation cover is simulated under the retreating ice sheet? How different are the properties of this land cover with respect to a glaciated surface?*

We have inserted, "When the ice-sheet retreats, the newly exposed land is assigned the properties of bare soil, including a low snow-free albedo; its properties do not subsequently change because vegetation dynamics are not included in the model."

**(8)** *The spinup is made under 1980-1999 MIROC (ocean) boundary conditions, but scenario climates from other three models are applied without going through the historical period of the corresponding model. For instance, in Figure2a, the change in SMB is referred to historical MIROC, but in the text the historical value for a different*

[Figure]

*model is given for this change in SMB. In Figure 3 historical HadCM3 SMB is depicted in column a, while the text in Section 6 indicates that this climate is not used.*

We think the comment about Figure 2a refers to the mistake we have corrected following the referee's comment on line 215. The caption for Figure 3a has been corrected to read "initial state with HadGEM2-ES historical climate"; the referee is correct that this is not a spun-up steady state of the ice-sheet, although not far from being so. Apologies—we do not understand the comment about Section 6.

**Abstract** *The abstract is difficult to understand without having read the body text first. Much context is missing on the numerous complex statements. I suggest to give more context and/or reduce/generalize the conclusions.*

As suggested in following comments, we have deleted some sentences which were evidently too complex or needed too much explanation.

**L7** *clarify meaning of "initially", and of "warming".*

We have deleted the phrase containing "initial". We think "global warming" is a well-understood phrase.

**L7** *A bit of introduction on the simulation design (e.g. "steady state", use of BC from four different models) could be helpful.*

We believe that "steady state" is a familiar concept. We don't think there is enough space to say more about the experimental design.

**L8** *"for all RCP8.5 climates"—clarify that this means for a steady state climate corresponding to 2081-2100, and not RCP/ECP8.5 up to 2300 as* e.g. *in Aschwanden* et al.*, 2019, Vizcaino* et al.*, 2015*

We have deleted this sentence, for brevity and because we have inserted, "For warming exceeding 3 K, the contribution to GMSLR exceeds 5 m", in response to comments by referees 1 and 2.

**L9** *"if recent climate were maintained": this is a conclusion not from the simulations, but from empirical relationships obtained from the simulations.*

Yes, it is an estimate obtained from a fairly good linear fit to the results, so it is in effect an interpolation. We don't think that is essential to include in the abstract. Also note that we have corrected the range from "1.5–2.5" to "0.5–2.5".

**L11,L12** *"The dominant effect is reduction of area": effect on what? The statement is very cryptic, why is the area important? "The geographical variation of SMB must be taken into account". This seems to imply that this is not taken into account in previous work, but one could argue that simulation of ablation and accumulation area is a geographical variation already . . . . Please specify further "geographical variation of SMB"*

We have deleted the sentence; these comments indicate that it's too complicated for the abstract.

**L14** *"if late twentieth-century climate is restored . . . the ice sheet will not regrow to its present extent"—this would be straightforward after reading line 9, but this line 14 is based directly on the model simulations, and not on the empirical relationships obtained . . . the mixing of direct and indirect results is a bit confusing.*

We don't share the referee's concern about this. As stated above, the estimate of the final mass for present SMB is an interpolation of the model results, which is not really "indirect" in our opinion. The other statements about the final mass simply describe model results.

**L15** *"owing to such effects": which effects? The cloud and snowfall effects were just said to increase the SMB, so they would help to re-grow the ice sheet?*

Thanks for pointing to this possible confusion. The "effects" we meant were regional climate change, but in this case they have the opposite consequence, as the referee says. We have rewritten the sentence, also following a similar comment by referee 2.

[Figure]

**L71** *"Because of the elevation and albedo feedbacks ... the present ice-sheet could not be regenerated"—Is a decrease of albedo "irreversible"?*

The loss of the ice-sheet might be (at least partly) irreversible, because the albedo and elevation are lower than in the present state, making the SMB more negative. The albedo change itself could be reversed if the ice-sheet readvanced, of course. We have rewritten this sentence as follows and hope this avoids the misunderstanding: "Even after $CO_2$ fell and global climate returned to pre-industrial, it might not be possible to regenerate the ice-sheet, because of greater ablation or reduced snowfall due to lower elevation and albedo in deglaciated regions."

**L123–124** *Muntjwerf* et al. *(GRL, 2020) also use this approach with same number of elevation classes for CESM2.*

Thank you for this reference, which we have inserted, along with Lipscomb *et al.* (2013).

**L126** *Sellevold* et al. *(TC, 2019) discusses the sensitivity of the "tiles" downscaling method to lapse rate choice.*

Thanks for this reference, which we have inserted at the point just below where we discuss the lapse rate, noting that $6 \, \text{K} \, \text{km}^{-1}$ gives the best SMB gradient in their study.

**L129–L120** *Which gradients are these, can you specify?*

The gradient of downwelling longwave radiation is $3.6 \, \text{W} \, \text{m}^{-2} \, \text{K}^{-1} \, \text{km}^{-1}$. The specific humidity gradient is not a constant because of the strong dependence of saturation specific humidity on air temperature. Since these are described by Smith *et al.*, we think that they are adequately described here by our phrase "consistent with the prescribed lapse rate".

**L134, L137** *"There is an uncertain parameter ...", please specify. Since Smith* et al. *is in preparation, could you give here more detail on the albedo modeling?*

We would rather not repeat too much, and think that for the purpose of this paper the

summary here is adequate, since Smith *et al.* is now available online and gives further information.

**L151–L157** *How is sliding parameterized in the ice sheet model?*

We have inserted a comment that we run the model with no sliding.

**L163** *"we run 10 years", do you mean, each atmospheric year is used 10 times, then next atmospheric year is run, etc?*

Yes, that's right. We have rewritten this sentence in the hope of making it clearer, thus: "Therefore, after each AGCM year, the ice-sheet model runs for ten years with the resulting SMB field, depending on the assumption that the elevation–SMB feedback will be negligible for changes in topography that occur within that decade, before the AGCM runs again."

**L165** *"We have verified": would it be possible to show some proof of this,* e.g. *a figure in the supplementary information?*

We have added another appendix (now Appendix A) to show both this and the effect of using monthly BCs including interannual variability instead of climatological monthly means.

**L176** *Title "3.3" can be more precise, maybe specify "(Evaluation of) Simulated 1980-1999 surface mass balance"?*

We have made it "Simulated surface mass balance for recent climate", to be more informative, and to contrast with the following sections on warmer climates. It's correct that the climate is nominally 1980–1999, but the climate data is from AOGCM simulations, which do not simulate real-world unforced interannual variability, and the FAMOUS–ice SMB is compared with RCM SMB—observational data is not involved.

**L193** *"Similar" is perhaps subjective. Please discuss similarities/differences.*

On further analysis, we found that ELA is generally greater in FAMOUS–ice. Smith

Interactive
comment

*et al.* have a paragraph about this comparison (their p13). We have replaced this sentence with their summary and a reference "The equilibrium line (black contour) is generally a little higher and further inland in FAMOUS-ice (see Smith *et al.* for details)."

**Table 1** *Last sentence is unclear (20, 30 or 100 years, "second group"?)*

We have rewritten this to avoid the unclear mention of "groups" and provide further information, thus, "The RCM time-means use 20 years of data, while we use 100 years for the FAMOUS–ice MIROC5 1980–1999 simulations, which supply our initial steady states, and 30 for other FAMOUS–ice simulations, which are transient states."

**Table 1** *The SMB for RCP2.6 is very similar to historical. This makes sense as only the 2080–2099 (steady-state) climate is used, as compared to other SMB estimates with evolving ice sheet topography where the full 21st century climate for RCP2.6 is applied. Maybe good to add some cautionary text to avoid misleading comparisons?*

The SMB change under RCP2.6 is quite small because the climate change under that scenario is fairly small. The MIROC5 RCP2.6 results in the table are from the experiments marked as squares near 1.0 K and between $-100$ and 0 Gt yr$^{-1}$ in Figure 2a; CanESM2 and HadGEM2-ES have similar temperature change for RCP2.6, while NorESM1-M, shown as crosses, has about 0.5 K. For RCP2.6, MIROC5 gives smaller SMB change than the other three AOGCMs, but it is within or just outside the AR5 uncertainty indicated with dashed lines. For RCP8.5, MIROC5 is in the middle of the range (the squares at around 3.5 K).

We chose to report the MIROC5 medium-albedo results for comparison with the 1980–1999 climates in the table, but we agree with the comment that this choice could lead to an inaccurate impression of the effect of climate change, especially for RCP2.6, where the systematic uncertainty is proportionately large. Therefore we have included in the table the average of the results of the four AOGCMs for the medium albedo under each scenario, and stated the AOGCM-average SMB change in the text.

We think that the neglect of topography change during the 21st century is compara-
tively unimportant. At the end of Section 4.1, we report that the elevation feedback
is about 20% of the SMB change in the second century for initial perturbations more
negative than $-100$ Gt yr$^{-1}$; for the first century with smaller initial perturbations it is not
distinguishable from statistical uncertainty in our results. Edwards *et al.* (2014) give a
best estimate of 4.3% by 2100 under larger climate change than RCP2.6. We have
remarked on this in the text.

**Table 2** *Why 402 ppmv for the historical period, isn't it too high? The mean concentra-*
*tion was lower than that in 1980-1999.*

The $CO_2$ concentration is "equivalent $CO_2$", used to represent all forcings. We treated
1980–1999 as "present day". The AR5 median assessment of the net anthropogenic
ERF in 2011 was 2.3 W m$^{-2}$, with a likely range of 1.1–3.3 W m$^{-2}$. Because the dif-
ference between this and the nominal forcing of 2.6 W m$^{-2}$ under RCP2.6 at 2100 is
small compared with the large systematic uncertainty in present-day forcing, because
the forcing is anyway much less important than the SST boundary conditions, and be-
cause our simulations are intended more as indicative than as realistic scenarios, we
decided for simplicity to use the same concentration for historical and RCP2.6 simula-
tions. We have added further comments in the caption of Table 2 about this.

**L215** *In the legend of Figure 2 it says delta_SMB is referred to steady state under*
*historical MIROC5, but here the HadGEM2-ES value (+307) is given, which one is*
*correct?*

Thanks for noticing this. We have modified the text to give the correct $\Delta$SMB of
$-1066$ Gt yr$^{-1}$ (rather than $-1063$) relative to the MIROC5 historical climate with low
albedo (rather than the HadGEM2-ES historical climate), and removed the SMB of
$+307$ Gt yr$^{-1}$ from the text to avoid confusion.

**L220** *10% larger—I get 14%, am I missing something? (0.67*1.50/0.88)*

To be precise, it is 13% ($= 797/708$, the numbers shown in Figs 3a4 and 3b4, which we refer to the text), or 10% to one significant figure. We have inserted $\sim$ to indicate the rounding, and also with the 50% in the text sentence (which is 49%, to be more precise), and we have rounded the 88% and 67% for consistency.

**L238** *"becomes gradually more positive" → increases*

We think "increases" is ambiguous when discussing a negative number, as it might mean "increases in magnitude", which is the opposite of "becomes more positive".

**L269** *"there are such states" : Could you add detail on those? It is difficult to map them from Figure 2d to Figure 2b to follow the SMB evolution.*

This comment suggests a simpler way to make the point, which is that all the final steady states have positive SMB and non-zero $M$, even though many of the trajectories start with negative SMB in Figure 2b. We have changed the text accordingly.

**L292** *smaller → lower*

We have inserted "in magnitude" to clarify the meeting.

**L299** *"more negative", "4 times more"—confusing, please give values*

The actual values are all negative and the point is mainly qualitative; we think the confusion is about what a larger negative number means. We have inserted the numbers in parenthesis with the Figure references to clarify the comparisons being made.

**L307–308** *precipitation contours are difficult to read or labels are absent*

We have now distinguished the 0.5 and 1.0 contour lines by linestyle.

**L368, L371** *"All but one": are you explaining this "one" somewhere else?*

Yes. These exceptions are discussed in Section 6.3. We have inserted a comment in parenthesis.

**L398** *"we suppose this dome might regrow in time": it seems it does not regrow in*

*20,000 years, when do you expect it will start to regrow?*

Fair comment. We can't rule it out, and maybe some unforced variability might stimulate it, but we have no evidence that it will regrow, so we have deleted this remark. We have also deleted the similar speculation about the regrowth of the southern dome in the EWN state with low albedo.

**L411** *"difference" in what? "infinitesimal"—what does this mean? Please quantify.*

We have rewritten this sentence to make it less abstract.

**L435–439** *Why 2,500 years at 0.7 mm/yr and 1,700 at more than double the rate (2 mm/yr)? How have you done these calculations? Which GrIS mass are you considering as present-day mass and as NON steady-state mass?*

We have now stated all the numbers to the nearest 100 years (previously we had rounded the first two to the nearest 500 years). They are $(7.4 - 4.0)/0.7 = 4857 \simeq 4900$, $(7.4 - 5.5)/0.7 = 2714 \simeq 2700$, $(7.4 - 4.0)/2.0 = 1700$ years and $(7.4 - 4.0)/6 = 567 \simeq 600$ years. The no-north (NON) steady-state mass of 5.5 m SLE is stated in the previous paragraph. The GrIS present-day mass of 7.4 m SLE is stated in Section 1.2.

**L452** *"outweighed": in which sense?*

We don't understand this comment. You could say that the positive feedbacks are overwhelmed by the negative feedbacks. That is the sense, but "overwhelmed" sounds too strong.

**L453** *"Snowfall": do you mean ice-sheet-integrated?*

Yes—clarified.

**L466** *"If a climate (...)". There is a jump here. The results now are based on the empirical relationships, but not directly on the simulations here as the ice sheet is relatively stable under 1980–1999 forcing.*

Here we refer to the recently observed imbalance, rather than the 1980–1999 steady state. We have rephrased it.

**L479–481** *See* **Low resolution**, *above*

**L485** *"sketchy" has a negative meaning in informal American English, maybe replace by other adjective ("gross")*

We have deleted the adjective.

**Figure 1** *White contour in b), is this modelled or observed "ice margin"?*

The model ice margin coicides with the observed one; we have inserted an explanation of this in the text (in the first paragraph of Section 2.2, formerly 3.2).

**Figure 1** *The ELA contour of c) makes a strange shape in the NW, any idea why?*

The horizontal SMB gradient is small in this region and the bedrock topography has an inlet, but we do have a clear explanation.

**Figure 2** *It is difficult to read precipitation from the last row of figures. Only one contour line is labeled, there are very few others, and no line interval is given.*

We believe that this comment refers to Figure 3. It is similar to the referee's comment on L307–308. We have now distinguished the 0.5 and 1.0 contour lines by linestyle.

**Figure 4** *Legend text and keys are confusing. My understanding is that the line color indicates albedo choice. "[Timeseries of Greenland ice-sheet mass with constant climates and FAMOUS–ice albedo] indicated by the line colors according to the line key of (c)" would hint to the colors indicating climates, but it is actually albedo?*

This is a misinterpretation of what we intended the sentence to mean. We have broken it into two sentences to avoid the misinterpretation, thus: "Timeseries of Greenland ice-sheet mass with constant climates. The FAMOUS–ice albedo is indicated by the line colors . . . ".

[Figure]

**Figure 4** *I would take the line key from (c) and put it in a common space as it applies to all panels. Same for the symbols in (c).*

Thanks for the suggestion, now implemented.

**Figure 4** *The orange dotted line corresponds to offline simulation with low albedo, shouldn't it be red dotted for consistency? Otherwise, add it to the key for lines, with a "Low albedo, offline run"*

Yes, it was orange by mistake, and is now red.

**Figure 4** *Panel b legend: "the circles indicate transient and final sites" is not clear. Having circles in the key is confusing, as only the colors are used, and those correspond to the symbols in the key of (c).*

Yes, good point. We have changed the key to the final symbol colours in (b) and hope the new version is clearer.

**Referee 2 (Alexander Robinson)**

**Threshold or no threshold.** *See **Threshold**, above*

**Section 2 on "Conceptual basis for the existence of a threshold warming."** *I don't quite see how this section adds value to the manuscript, as it is currently framed, especially since later it is stated that the "the conceptual basis for its existence is incorrect". It feels somewhat like a straw-man argument. The simple equations described in Section 2 are useful for conceptualizing the possibility of a runaway feedback leading to the complete melting of an ice sheet. But I think it is by now clear to the community that an ice sheet like Greenland is a large, complex system with processes coupled to atmospheric circulation and a wide range of acting timescales. It is clear, for example, that $\Delta s$, $A$ and $f(\Delta M)$ will all change over time. Therefore, I would suggest to the authors that, rather than framing this as the current paradigm that should be rejected, it would be more valuable to highlight, conceptually, what could happen when some*

of those terms vary (i.e., when $A$ becomes smaller, but $\Delta s$ increases), or to expand $f(\Delta M)$ into the multiple contributions that may exist (like $f_{\text{albedo}}$, which is most often positive in a warming climate, and $f_{\text{cloudiness}}$, which is found here to be an important negative feedback). Because I think the authors would agree that, ignoring possible climate feedbacks like cloudiness for a moment, as done in the uncoupled experiment, the theoretical basis for a threshold for ice sheet retreat (i.e, the small ice cap instability) still applies here. It is just mitigated by additional feedbacks/factors that are not accounted for in this simple equation. Along the same lines, I don't think it makes sense to summarize the study of Robinson et al. (2012) at the end of this section as estimating $\Delta M$. In that study, as in this one, a fully coupled climate–ice-sheet system is simulated with a dynamically evolving ice sheet and topography. The results of such experiments allow later comparison with expectations from this conceptual framework, but this equation is not used at all for any quantitative analysis.

The idea of an abrupt threshold is familiar because it has been demonstrated in previous literature, but it is often not explained how it comes about. The intention of this section is simply to do that. It isn't intended to be a "straw man" in the sense of misleading anyone, but it is indeed an simplification. We have demoted it to become a subsection (1.3), now entitled "Discussion of the threshold warming", just after the idea of a threshold is introduced, where the explanation may help most. In revising the text, we have tried to make its intention clear, and have rewritten much of it. We hope that referee 2 finds that the study of Robinson *et al.* is now better represented. We have taken up the implied suggestion to use the same framework in the later discussion (formerly Section 5, now Section 3.5) to interpret the results.

**(1)** *I am very surprised to see an SIA model applied here, and at only 20km resolution. At a minimum, some justification of this relatively low grid resolution should be made (I would not expect computational cost to be an issue for such a model in this framework). More importantly, the authors should acknowledge and discuss the possible impact of a lack of fast ice dynamics in their simulations. For example, is basal sliding param-*

*eterized in some way, or is no basal sliding allowed? If it is not, the model is likely underestimating the dynamic adjustment of the ice sheet to the area retreat, which is an important positive feedback on ice decline on these timescales.*

Computational cost is actually a consideration. The ice-sheet model is 20% is the cost of the coupled model. If we doubled the resolution of Glimmer, that would presumably increase its cost fourfold and double the cost of the coupled model and therefore the wallclock time of these experiments, which took many months to run on the resources available. The model is fast, but these are very long experiments!

There is no basal sliding in the model. We are not simulating ice-streams or rapid ice dynamics. We have inserted comments to this effect in the description of the model. To simulate these phenomena properly would require much higher resolution in some regions as well as higher-order dynamics, with much greater cost. We agree that omitting rapid ice-sheet dynamics means that the rate of ice-loss will be underestimated, but the aim of our experiments, with their simple scenarios of constant late 21st-century climate, is to investigate the steady state. We have added this explanation in the model description and as a caveat in the conclusions.

**(2)** *See* **Low resolution***, above*

**(3)** *It would also be valuable to see a figure showing the forcing applied to the model. For example, how do the present-day SST fields compare to reanalysis or observed SSTs? What are the future patterns of warming? Also, it generally seems that the simulations forced by NorESM1-M stand apart from the other two with lower ice volumes predicted for the same level of global warming. Is this reflected in the SST warming patterns in some way?*

We chose these four AOGCMs because of their previously having been assessed as relatively satisfactory for simulation of Greenland regional climate for our reference period (1980–1999) by Fettweis *et al.* (2013) and van Angelen *et al.* (2013). This rationale and these references are given in the manuscript; we have now drawn attention

to them for information about the climate evaluation. We have included further plots of global-mean and Greenland regional mean surface air temperature range (in Appendix C) showing that NorESM1-M has the smallest warming in general.

**(4)** *Finally, in terms of style, I find that the use of abbreviations for different variables throughout the text makes the manuscript harder to follow. For example, on L30, the phrase "The increase in $D$ is probably the ice-dynamical response ..." would be more straightforward replacing "$D$" with "discharge". Perhaps the authors could consider only using the variable abbreviations ($P$, $D$, $R$, $M$, etc.) when the text is related to specific equations that use them, and otherwise use the actual names in sentences. Some other abbreviations could be avoided all together (BCs, GMSLR, etc.).*

Thanks for drawing attention to this. We had used the symbols for terms in the mass balance especially and unnecessarily in the introduction and the two conceptual discussions. We have now replaced them with words except where they are needed in equations, as suggested. However, we have kept $M$ because it is widespread throughout the text (about 30 occurrences), and quite a lot shorter than "[Greenland] ice-sheet mass", and $\Delta$SAT for change in global-mean surface air temperature. We have also kept "SMB" (over 100 occurrences), "BC" (19 occurrences), "GMSLR" (26 occurrences) and "SLE" (38 occurrences), which are all fairly well-known abbreviations for rather long phrases.

**L11-12** *"This is because the dominant effect is reduction of area, not reduction of surface altitude, and the geographical variation of SMB must be taken into account." This sentence could be more precise. In previous work geographical variation of SMB was also considered, even if in a simpler way. Nor does it seem that the dominant effect is the reduction in area. Rather, it seems that changes in atmospheric circulation act to mitigate the warming via increased cloudiness. The next sentence is already clearer, so I would suggest deleting this one.*

We have deleted it.

**L15** *"owing to such effects." This reference is not very clear, as increased cloudiness and precipitation would, in principle, help the ice sheet regrow. Consider rephrasing, or simply removing.*

Thanks for this point. We agree that it was confusing. We have rewritten the sentence, also following a similar comment by referee 1.

∼**L118, Section 3.1** *Other boundary conditions aside from topography and SSTs should be explicitly mentioned here. Are greenhouse gas concentrations applied in FAMOUS–ice to be consistent with the applied SST fields, for example?*

The BCs are introduced three paragraphs later in the same section (at line 143 of the submitted manuscript). Yes, the radiative forcing is consistent with the climate of the BCs. In the later paragraph we have now mentioned this, and made reference to Table 2 and Section 4.

**Fig. 3** *This is an important figure, but feels a bit busy. Perhaps the color bars of each row could be placed vertically on the side? This would clean it up a bit and make more room for the panels themselves.*

We sympathise with this comment, and considered various designs, but couldn't find a better one for fitting this much information into one figure. If it were split into more than one figure, it would be harder to compare the corresponding cases. There is a different colour bar for each row, and a single row isn't high enough for a colour bar on its side. Without the colour bars between the rows, the individual panel could indeed be enlarged, but only in the north–south direction, which would make them look unlike Greenland as we know it.

**Fig. 4** *This figure is hard to follow, as there is a lot of information. I would suggest revising colors and symbols to provide more clarity. For example, I think the colors for different forcing scenarios should be substantially different from the colors delineating model versions (low, medium, high albedo).*

As suggested, we have changed the colours for the scenario to be a different set from the colours for the albedo, and hope that this is less confusing. Also, we have adopted the suggestion of Referee 1 to put the key for line colours and symbols outside the panels, since it is common.

**L269–270** *A stable but diminished ice sheet is consistent with previous work. Robinson et al. (2012) found that 10% of the ice sheet remained even above the tipping point for large scale melting (see state E3 shown in Fig. 4 of that reference for an approximate picture). It is clear that at some point retreat of the ice sheet to high elevation zones may lead to restabilization. I suggest reframing here.*

We have deleted the sentence here. We have made a remark along these lines in Section 3.5 (formerly Section 5).

**L337–345** *Consider reframing title of this section and first paragraph along the lines of earlier comments.*

We have demoted Section 5 to a subsection (now 3.5), entitled "Discussion of reduced steady states", and rewritten it following the comments by referees 1 and 2 under the heading **Threshold**.

**L387–388, L395, etc.** *I would remove the terms WOWS and NON, as they do not help comprehension beyond the already defined EWNS terms, and are used rather rarely in any case.*

We have removed "WOWS", which was referred to only once after its definition. We have replaced "NON" with "no-north", which may be clearer, and is useful because it stands for a group of three configurations and is referred to eight times.

**L411–412** *I would point out that this seems to be an example of a tipping point being activated. This is, of course, not starting from present day, and so is not the same as what has been discussed until now. But it does show that the mechanisms for triggering self-sustained decline are present in the system.*

We have inserted, "Due to these feedbacks, there is more than one steady state for the given BCs." We would rather not use the phrase "tipping point", because that's popularly used for unstable transitions in the opposite direction (as in Section 1.3), so its meaning here might not be clear.

**L458** *Greenland or global warming → regional or global warming*

We have put "regional Greenland or global warming".

**Referee 3 (Xavier Fettweis)**

**(1)** *In addition to Fig1, it should be interesting to show the differences/biases with the "reference". In the legend, is it MAR forced by MIROC5? or ERA?*

We have included difference maps in Figure 1. MAR is forced by MIROC5, so the results of the two models are comparable.

**(2), Lines 183–187:** *The near-surface climate from RCMs forced by a GCM cannot be compared directly with the near-surface climate from the forcing GCM as RCMs simulate their own boundary layer climate and are even able to correct near-surface GCM biases. As FAMOUS–ice is forced by near-surface climate from GCM, it is normal that there are differences with the RCMs simulations. The RCM are only sensitive to the free atmosphere climate from GCMs. Therefore, this section should be a bit rephrased to explain this issue.*

Yes, we agree. We have changed this sentence to say, "A similarly large spread in SMB arises from the choice of Greenland model (FAMOUS–ice, MAR or RACMO), both because they simulate somewhat different regional climate in the free atmosphere and over land when given climate BCs from the same AOGCM, and because they have different SMB schemes."

**(3), Lines 314–317:** *See **Low resolution**, above*

Please also note the supplement to this comment:

https://tc.copernicus.org/preprints/tc-2020-89/tc-2020-89-AC1-supplement.pdf

[Figure]

[Figure]

**Fig. 1.** (a) is Figure 1C of Levermann et al. (2013). (b) is Figure 2c of our manuscript.